# Proteasomal down-regulation of the proapoptotic MST2 pathway contributes to BRAF inhibitor resistance in melanoma

David Romano[1], Lucía García-Gutiérrez[1], Nourhan Aboud[1], David J Duffy[1,4], Keith T Flaherty[3], Dennie T Frederick[3], Walter Kolch[1,2], David Matallanas[1]

**The RAS-RAF-MEK-ERK pathway is hyperactivated in most malignant melanomas, and mutations in BRAF or NRAS account for most of these cases. BRAF inhibitors (BRAFi) are highly efficient for treating patients with BRAF$^{V600E}$ mutations, but tumours frequently acquire resistance within a few months. Multiple resistance mechanisms have been identified, due to mutations or network adaptations that revive ERK signalling. We have previously shown that RAF proteins inhibit the MST2 proapoptotic pathway in a kinase-independent fashion. Here, we have investigated the role of the MST2 pathway in mediating resistance to BRAFi. We show that the BRAF$^{V600E}$ mutant protein, but not the wild-type BRAF protein, binds to MST2 inhibiting its proapoptotic signalling. Down-regulation of MST2 reduces BRAFi-induced apoptosis. In BRAFi-resistant cell lines, MST2 pathway proteins are down-regulated by ubiquitination and subsequent proteasomal degradation rendering cells refractory to MST2 pathway–induced apoptosis. Restoration of apoptosis can be achieved by increasing MST2 pathway protein expression using proteasome inhibitors. In summary, we show that the MST2 pathway plays a role in the acquisition of BRAFi resistance in melanoma.**

## Introduction

Metastatic melanoma is the most aggressive form of skin cancer, and its incidence is increasing worldwide, in particular in Western countries (Rigel, 2010; Feigelson et al, 2019; Davey et al, 2021). This tumour type is characterised by a high frequency of genetic mutations within the RAS-RAF-MEK-ERK pathway, with mutations in *BRAF*, *NRAS* and *NF1* as the main drivers of transformation (Curtin et al, 2005). The development of targeted therapies to target the transforming effects of mutations in these genes in melanocytes

has been the focus of intense research. This led to the development of BRAF inhibitors (BRAFi), which are extremely effective in melanoma patients with BRAF$^{V600E}$ mutations. The combination of BRAFi with MEK inhibitors (MEKi) further improved efficacy and duration of the response. However, most of these patients will develop resistance to BRAFi or BRAFi + MEKi combination therapies within a year (Arozarena & Wellbrock, 2017; Rossi et al, 2019). The mechanisms of resistance are pleiotropic and not fully characterised. In general, they result in the re-activation of ERK signalling and include the paradoxical activation of wild-type (wt) RAF isoforms, secondary mutations and genetic and epigenetic changes that result in a rewiring of signalling networks (Arozarena & Wellbrock, 2017; Rossi et al, 2019). In addition, reactivation of the PI3K/AKT pathway and up-regulation of anti-apoptotic signals also can convey BRAFi resistance (Perna et al, 2015; Serasinghe et al, 2015).

The Hippo/MST2 pathway (here called MST2 pathway) plays an important role in the regulation of proliferation, organ size and cell death (Pan, 2010; Fallahi et al, 2016). Originally discovered in *Drosophila melanogaster*, the pathway has functionally diversified in mammals, where it can drive both cell proliferation and cell death (Romano et al, 2014a; Fallahi et al, 2016). In mammals its core element is a kinase module consisting of MST1/2 which phosphorylates and activates LATS1/2 kinases with multiple substrates including the YAP transcription regulator, which is a main effector of the canonical Hippo pathway discovered in *D. melanogaster*. However, the mammalian MST1/2 can receive inputs from a variety of upstream cues and transmit signals via various downstream effectors in addition to YAP. The pathway's involvement in the regulation of apoptosis is stimulated by the FAS death receptor, by RASSF1A, and mutant KRAS (O'Neill et al, 2004; Matallanas et al, 2007; Matallanas et al, 2011b). *RASSF1A* is a tumour suppressor gene, whose expression is frequently silenced in cancer (Richter et al, 2009). These proapoptotic signals are antagonized by RAF1, which binds to and inhibits MST2 (O'Neill et al, 2004). RASSF1A competes with RAF1 for MST2 binding causing the release of MST2 from RAF1,

[1]Systems Biology Ireland, School of Medicine, University College Dublin, Dublin, Ireland [2]Conway Institute of Biomolecular and Biomedical Research, University College Dublin, Dublin, Ireland [3]Massachusetts General Hospital, Boston, MA, USA [4]Department of Biology/Whitney Laboratory for Marine Bioscience, University of Florida, Gainesville, FL, USA

Correspondence: david.gomez@ucd.ie; walter.kolch@ucd.ie

allowing MST2 activation, binding to LATS1/2, and subsequent promotion of apoptosis. Interestingly, apoptosis can proceed through two routes. One is mediated by MST2-LATS1 signalling inducing the formation of a YAP-p73 transcriptional protein complex that promotes the expression of proapoptotic genes (Matallanas et al, 2007). The other pathway, also mediated by MST2-LATS1 is independent of YAP and leads to the stabilization of the p53 tumour suppressor protein (Matallanas et al, 2011b).

Different lines of evidence indicate that the deregulation of members of the MST2 pathway plays a role in the development of malignant melanoma. YAP1 has been proposed to behave as an oncogene in melanoma (Thompson, 2020). RASSF1A is commonly lost in melanoma patients because of DNA methylation (Reifenberger et al, 2004). LATS1 levels seem to be decreased through different mechanisms that include post-translational modification and long non-coding RNAs (Yuan et al, 2015; Han et al, 2021). These results suggest that loss of a functional MST2 pathway might be associated with melanoma development. However, the possible role of the MST2 pathway in the acquisition of resistance to RAFi has not been studied.

Here, we have investigated the association of MST2 pathway signalling and the rewiring of molecular networks that result in the acquisition of resistance to BRAFi in melanoma cell lines. We show that mutant BRAF$^{V600E}$ binds and inhibits MST2 preventing the activation of MST2-dependent apoptosis. In BRAFi resistant melanoma cells developed in our group, we show that LATS1 and MST2 expression is reduced because of ubiquitin ligase-dependent degradation. Treatment of resistant melanoma cells with proteasome inhibitors rescues MST2 and LATS1 expression and restores proapoptotic signalling. Finally, results from a small cohort of patients with resistance to BRAFi indicate that MST2 down-regulation might be associated with the acquisition of resistance in human melanoma.

# Results

### The MST2 pathway is inhibited by mutant BRAF in melanoma cells and the effect is rescued by BRAF inhibitors

The MST2 kinase is an important regulator of cellular growth and proliferation, and abundant evidence shows that deregulation of the MST2 signalling network is associated with cancer development (Pan, 2010; Fallahi et al, 2016). We have previously shown that MST2 mediates a proapoptotic signal that is inhibited by RAF kinases (O'Neill et al, 2004) and were interested in studying the possible role of the MST2 pathway in the response to RAFi used in the clinic to treat melanoma. Therefore, we treated a panel of melanoma cell lines that included the two main driving mutations in melanoma, BRAF$^{V600E}$ (A375, SK-Mel28, and WM-793 cells) and the NRAS$^{Q61R}$ (SK-Mel2 cells) point mutations with the BRAF inhibitor PLX4032 (vemurafenib) for 1 or 24 h. Using phosphorylation of the activating MST1/2 T180 residue as a read out for MST2 kinase activity, the results showed that PLX4032 caused a rapid and sustained activation of MST1/2 in the three BRAF-mutant cell lines, whereas no effect was observed in SK-Mel2 cells that do not respond to BRAF inhibitors (Fig 1A). Importantly, we observed that overexpression of wild-type BRAF and the dimerization defective

BRAF$^{R509H}$ mutant (Poulikakos et al, 2010) did not inhibit MST2 activity, whereas expression of the oncogenic mutants BRAF$^{V600E}$ and BRAF$^{V600E/R509H}$ in HeLa cells caused a complete inhibition of MST1/2 basal activation (Fig 1B). These results indicated that MST2 might be regulated by BRAF$^{V600E}$ in melanoma cells, and that this does not require BRAF dimerization.

The MST2 network crosstalks with the ERK pathway. MST2 can induce the phosphorylation of RAF1 at the inhibitory S259 site (Romano et al, 2014a), whereas both RAF1 and oncogenic BRAF can bind to and inhibit the core kinases MST2 and MST1, respectively (O'Neill et al, 2004; Matallanas et al, 2007; Lee et al, 2011). Therefore, we tested if mutant BRAF also binds MST2. Co-immunoprecipitation assays demonstrated that MST2 readily bound to mutant BRAF$^{V600E}$ in melanoma cells, whereas overexpressed wild-type BRAF only showed a weak interaction (Fig 1B). The binding of mutant BRAF$^{V600E}$ was confirmed using a BRAF$^{V600E}$ mutant specific antibody (Fig 1C). Reversing the IP by immunoprecipitating MST2 also detected a strong association of MST2 with BRAF in cell lines expressing the oncogenic mutant and no association in cells that express only wildtype BRAF cells. Interestingly, in all mutant BRAF cell lines we observed a clear decrease of interaction of MST2 and BRAF$^{V600E}$ when the cells were treated with PLX4032. To test if the disruption of this interaction is specific to PLX4032, we treated A375 cells with other BRAFi including two other type I ½ and two type II inhibitors (Cook & Cook, 2021). All of them decreased the MST2-BRAF interaction 24 h post-treatment, but to a lesser extent (Fig S1A). A concomitant increase of LATS1-YAP interaction becomes apparent after 1 h of treatment with BRAFi (Fig S1B). Taken together, these observations indicate that PLX4032 and other BRAFi may activate the MST2 pathway by releasing MST2 from BRAF$^{V600E}$ inhibitory binding. If this hypothesis is correct, down-regulation of MST2 expression in mutant BRAF$^{V600E}$ cells should prevent PLX4032 induced cell death. Indeed, using siRNA to knock down MST2 by ca. 70% in the three mutant BRAF$^{V600E}$ cell lines reduced PLX4032 induced apoptosis between 70% and 50% (Fig 1D).

These results show that mutant BRAF$^{V600E}$ interacts with MST2 and inhibits its proapoptotic signalling. Moreover, the results suggest that breaking up the interaction between mutant BRAF$^{V600E}$ and MST2, causing MST2 activation, is part of the mechanism of action of PLX4032.

### Acquisition of resistance to BRAFi causes the down-regulation of MST2 pathway proteins

Most melanoma patients develop BRAFi resistance within a year after initiation of treatment. This is due to the paradoxical activation of RAF isoforms or a rewiring of the signalling network that circumvents the RAF blockade to achieve ERK activation (Matallanas et al, 2011a; Arozarena & Wellbrock, 2019). In light of the observation that the MST2 pathway participates in mediating cell death in PLX4032-sensitive mutant BRAF melanoma cells, we wanted to know if this pathway is associated with the acquisition of resistance to BRAFi. To study this, we generated resistant (A375-R, SK-Mel28-R, and WM-793-R) cell lines from the A375, SK-Mel28, and WM-793 cells by growing the cells in media containing increasing amounts of PLX4030 for a period of 6 mo (Fig 2A). Cells that could sustain proliferation in the presence of 3 μM PLX4032 were considered

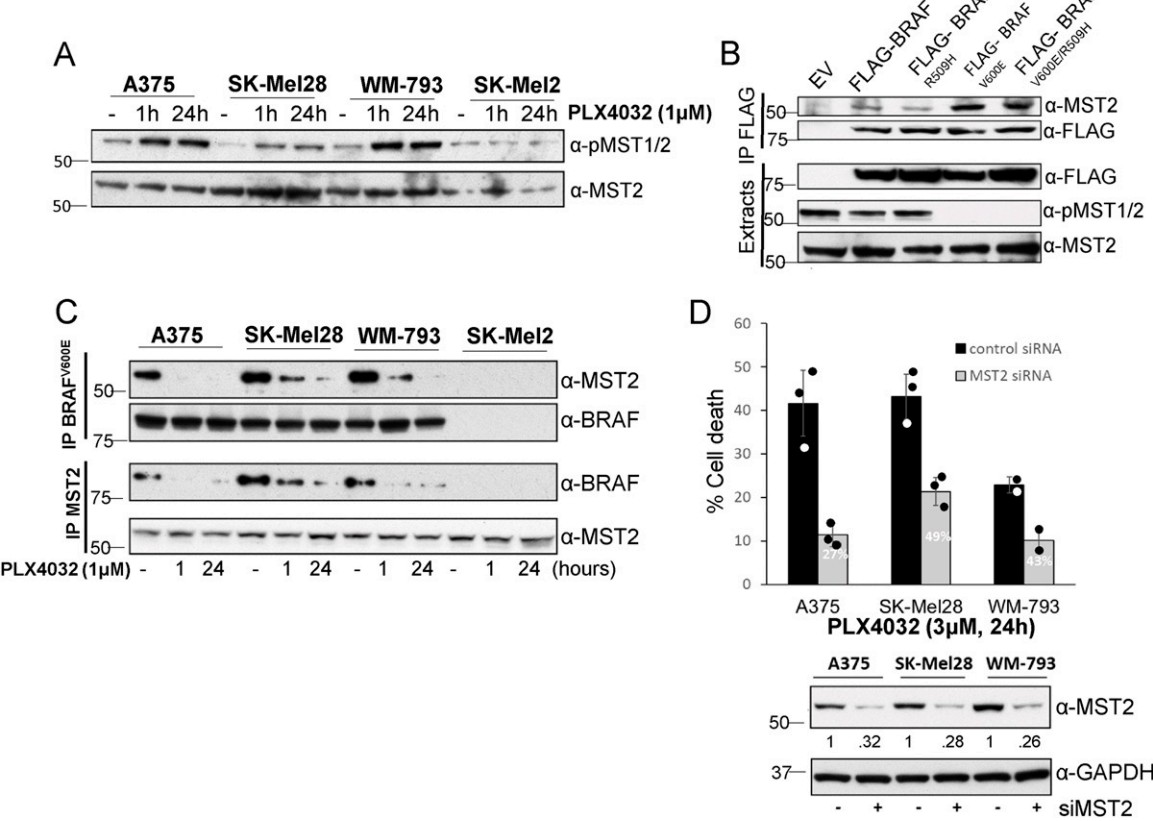

**Figure 1. BRAF<sup>V600E</sup> inhibits MST2 signalling.**

**(A)** A375, SK-Mel28, WM-793, and SK-Mel2 cell were treated with 1 μM PLX4032 for the indicated times. Cell extracts were analysed by immunoblotting with the indicated antibodies. **(B)** HeLa cells were transfected with pCDNA-FLAG (EV) or the different FLAG-BRAF constructs indicated. After 48 h, the cells were lysed and FLAG-BRAF protein complexes were immunoprecipitated using FLAG beads. BRAF-MST2 interaction and MST2 activation were determined by immunoblotting with the indicated antibodies. **(C)** A375, SK-Mel28, WM-793, and SK-Mel2 were treated with PLX4032 for the indicated times before MST2 and BRAF<sup>V600E</sup> were immunoprecipitated. Endogenous MST2 and BRAF interaction was monitored by blotting with the indicated antibodies. **(D)** Mutant BRAF cells A375, SK-Mel28, and WM-793 were transfected with non-targeting (control) or MST2 siRNA (50 nM). 24 h after transfection, the cells were treated with PLX4032 (3 μM) for 24 h and cell death was determined by assaying DNA fragmentation by FACS. Percent apoptosis in cells transfected with MST2 siRNA relative to control transfected cells is indicated. An aliquot of the cells was lysed and immunoblotted with the indicated antibodies to determine protein expression. The expression of MST2 normalized to the GAPDH loading control was determined by Image J scanning. Error bars show SD n = 3 for A375 and SK-Mel28; n = 2 WM-793. In all blots numbers indicate molecular weight (kD).
Source data are available for this figure.

resistant. We obtained several PLX4032-resistant clones. The resistant clones showed lower level of ERK activation than the parental cells. But, importantly, although treatment with PLX4030 causes a complete inhibition of ERK in parental cells, the resistant cells could maintain ERK phosphorylation (Fig 2A). These results are similar to other reports of BRAF inhibitor-resistant cells, confirming that the acquisition of resistance is due to a re-activation of ERK signalling (Lee et al, 2020). The resistant cells also had lost sensitivity to PLX4032-induced apoptosis (Fig 2B). As all clones showed similar responses in terms of ERK re-activation and resistance to apoptosis, a representative clone from each resistant cell line is shown (Fig. 2A–E) and was used to perform the follow-up experiments. Importantly, MST2 expression levels were down-regulated in all the resistant clones suggesting that losing the proapoptotic function of MST2 signalling is involved in the acquisition of resistance to BRAFi (Figs 2C and S2A and B). Therefore, we assessed the protein expression of LATS1, RASSF1A and YAP1 in the resistant cells, observing a strong decrease in their expression in

resistant cells (Fig 2C). The only exception was YAP1 expression in WM-793 cells, which did not change in the resistant cells. A375 and SK-Mel28 cells do not express RASSF1A, but RASSF1A is expressed in WM-793 cells and was reduced in their resistant variants. As expected from the reduction of protein abundances, the basal levels of MST2 and LATS1 activation were reduced in the resistant cells. Importantly, treatment with PLX4032 could not activate MST2 pathway kinases as it did in the parental cells. These data clearly indicate that both the abundance and activation of MST2 pathway proteins are down-regulated during the acquisition of RAFi resistance.

To further characterise the rewiring mechanisms occurring during the development of resistance to PLX4032 in our cell lines we used BRAFi and MEKi to probe the activation of the ERK pathway. All resistant cells showed an inhibition of ERK1/2 phosphorylation when the cells were treated with MEKi indicating that there was no secondary activating mutation of MEK in the resistant cell lines (Fig 2D). By contrast, PLX4032 failed to block ERK activation, presumably due to the induction of BRAF-RAF1 heterodimers (Fig S2C), which

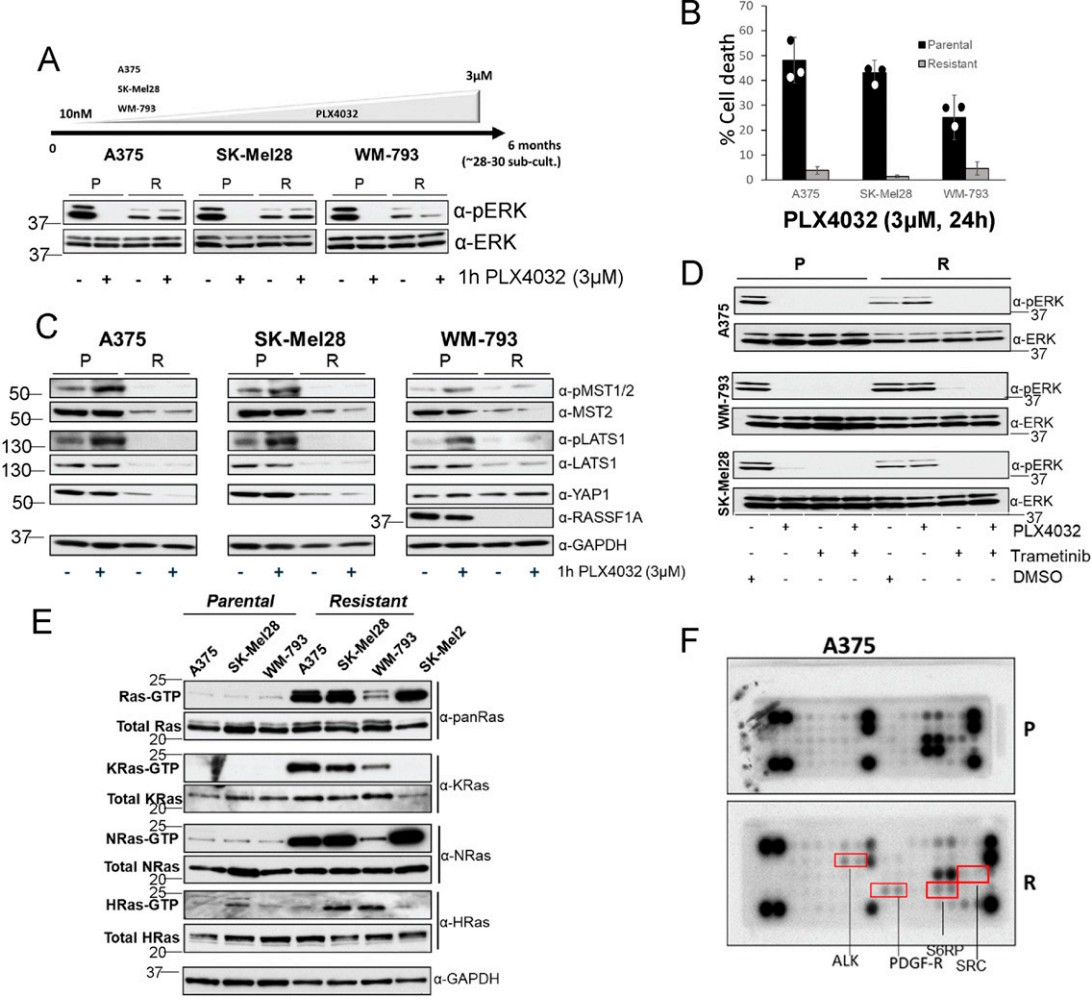

**Figure 2. Mutant BRAF melanoma cells resistant to PLX4032 cells rewire ERK and MST2 pathways.**
**(A)** A375, SK-Mel28, and WM-793 cell were grown in increasing concentration of PLX4032 for 6 mo. Parental (P) and resistant (R) cells grown without PLX4032 for 48 h were treated with PLX4032 (3 µM) for 1 h and ERK activation was determined by immunoblotting with the indicated antibodies. **(B)** Parental and resistant A375, SK-Mel28 and WM-793 cells were treated with PLX4032 (3 µM) for 24 h and DNA fragmentation was determined but PI staining using FACS. Error bars show SD, (n = 3). **(A, C)** Parental and resistant cells were treated as in (A) and the expression of MST2 pathway proteins and GAPDH as loading control were determined by immunoblotting with the indicated antibodies. **(D)** Parental and PLX4032 resistant cells were treated with PLX4032 (3 µM) and/or trametinib (10 nM) for 1 h. Cell extracts were immunoblotted with the indicated antibodies. **(E)** Lysates from serum deprived (16 h) parental and resistant cells were incubated with GST-RBD to determine the amount of RAS GTP. A lysate aliquot was used to determine total RAS expression. Expression of RAS isoforms was determined by immunoblotting with the indicated antibodies. **(F)** Parental (P) and resistant A375 (R) cells were lysed and cell extracts were incubated with PathScan RTK Signaling Antibody Arrays. Red boxes show ALK and PDGF-receptor spots. In all blots numbers indicate molecular weight (kD).
Source data are available for this figure.

have been shown to maintain ERK signalling or even cause paradoxical pathway activation (Arozarena & Wellbrock, 2017; Rossi et al, 2019). In parental cells, both PLX4032 and trametinib prevented ERK activation, whereas only trametinib inhibited ERK activation in PLX4032 resistant cells (Fig 2D). Treatment of A375 cells with PLX4032 and trametinib alone or in combination resulted in a slight activation of LATS1 in parental but not in resistant cells. Importantly, none of the treatments had any impact on the expression or measurable activation of LATS1 in the resistant cells indicating that the reduction of the MST2-LATS1 pathway in resistant cells is not mediated by ERK1/2 (Fig S2D). In addition, we could not detect the interaction between BRAF and MST2 in A375-R cells which may be due to the low level of expression of MST2 in these cells (Fig S2E).

Enhanced RAF dimerization is a main source of resistance to BRAFi in melanoma and is frequently caused by RAS activation (Johnson et al, 2015). Indeed, RAS activation was clearly increased in all three resistant cell lines, especially in A375 and SK-Mel28 resistant cells which showed similar levels of RAS activation as the mutant NRAS cell line SK-Mel2 (Fig 2E). Using isoform specific antibodies, we observed that A375 cells featured a strong activation of KRAS and NRAS but not HRAS. In SK-Mel28 resistant cells all RAS isoforms were hyperactivated with respect to the parental cells. Finally, in WM-793-R cells there was a weaker activation of NRAS compared with the other resistant cell lines, but KRAS and HRAS were clearly hyperactivated compared with the parental cell lines. This increase of RAS signalling could be caused by secondary

mutations in RAS isoforms. Indeed, this might be the case for the A375-R cell lines where we identified a mutation in NRAS, but we could not identify mutations in RAS genes in the other resistant cell lines. Thus, a more likely explanation is that RAS activation is caused by the activation of upstream regulators, which causes hyperactivation of several RAS isoforms. To test this, we used PathScan receptor tyrosine kinase (RTK) signalling antibody arrays. All cell lines had up-regulated RTK signalling. A375-R showed an increased phosphorylation of PDGF-R and ALK and a decrease of c-ABL and SRC phosphorylation. SK-Mel28-R showed an increase of IRS1 and SRC activation and lower ERK1/2 and S6RP phosphorylation (Figs 2F and S2F). Finally, WM-793-R had an increase of activation of IGF-1R, S6RP and the EGF-R effector SRC. These data indicate that the increase of activation of RAS isoforms may be caused by the activation of upstream receptor signalling in the resistant cells. The fact that the three cell lines showed activation of different receptors may explain why there is a differential activation of the RAS isoforms since we and other have shown that the isoforms are differentially regulated by upstream signals (Kiel et al, 2021).

In summary, the resistant cell lines that we have generated have acquired significant changes in the signalling machinery including diverse changes in RTK and RAS signalling, but similar changes in the ERK and MST2 pathways.

### MST2 pathway proteins are down-regulated by proteasome degradation in BRAFi resistant cells

These results indicated that a decrease of expression of MST2 pathway proteins is involved in BRAFi resistance. Therefore, we tried to identify the mechanisms behind this down-regulation in the BRAF[V600E], PLX4032 resistant cell lines.

Loss of expression of RASSF1A in A375 is due to DNA methylation of the gene promotor (Yi et al, 2011), and the expression of MST1/2 and LATS1/2 in cancer also can be regulated by DNA methylation (Fallahi et al, 2016). Therefore, we first tested whether the expression changes of core Hippo proteins were caused by decreased gene transcription. Measuring mRNA expression of MST1/2, LATS1/2 and YAP1 by RT-PCR showed equal or up-regulated expression in all resistant cell lines (Fig 3A). This makes it unlikely that the reduction of protein expression is due to gene silencing or a reduction in gene expression.

Overexpression of a mRNA can be a compensatory mechanism to counterbalance the enhanced degradation of the cognate protein (Prelich, 2012). To test this possibility, we treated the cells with the proteasome inhibitors bortezomib and Proteasome inhibitor 1. Whereas in parental cells these inhibitors had little effect on the expression levels of MST2 pathway proteins (Fig S3), both inhibitors consistently and rapidly increased the expression of MST2, LATS1 and YAP1 in all three resistant cell lines (Fig 3B). These data indicated that the reduction in MST2, LATS1, and YAP1 proteins expression in PLX4032 resistant cells is due to proteasomal degradation.

### Increased ubiquitination promotes MST2 and LATS1 proteasomal degradation in BRAFi resistant cell lines

Proteasomal degradation usually is preceded by ubiquitination, which directs the target protein to the proteasome (Kornitzer & Ciechanover, 2000). To test if there was an increase of ubiquitination of MST2 and

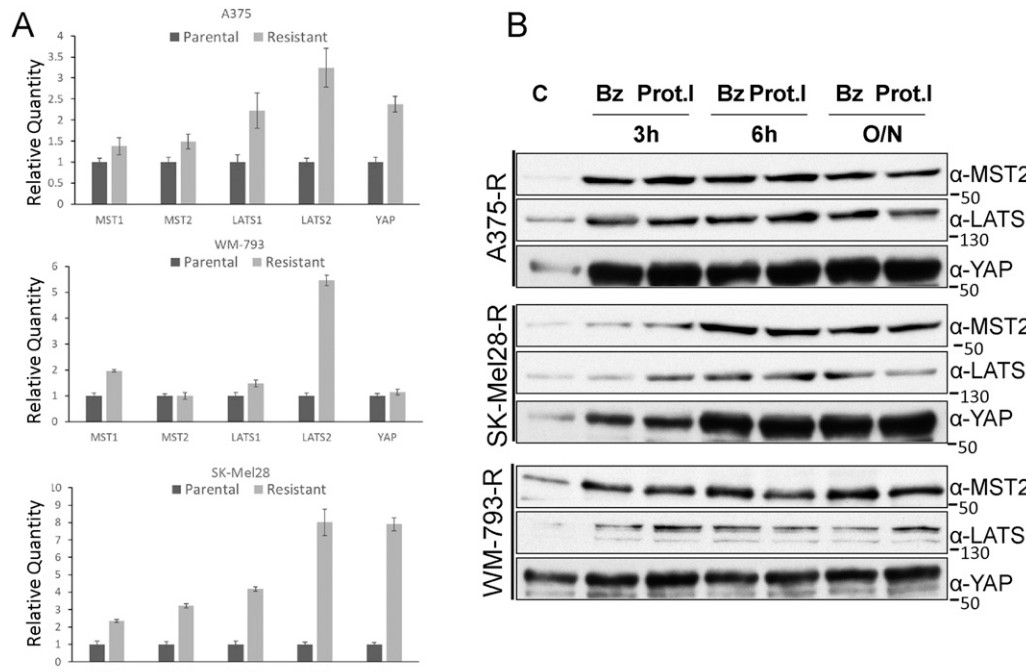

**Figure 3. MST2 pathway proteins are down-regulated by proteasomal degradation in PLX4032 resistant cells.**
**(A)** RNA was extracted from parental and resistant A375, SK-Mel28, and WM-793 cells and the level of expression of the indicated RNAs was determined by qRT-PCR. Error bars show SD for three measurements. **(B, C)** Resistant A375, SK-Mel28 and WM-793 cells (C) were treated with bortezomib (Bz, 50 nM) or Proteasome inhibitor I (Prot.I, 10 μM) for the indicated times (O/N, overnight). Cell lysates were blotted with the indicated antibodies. In all blots numbers indicate molecular weight (kD). Source data are available for this figure.

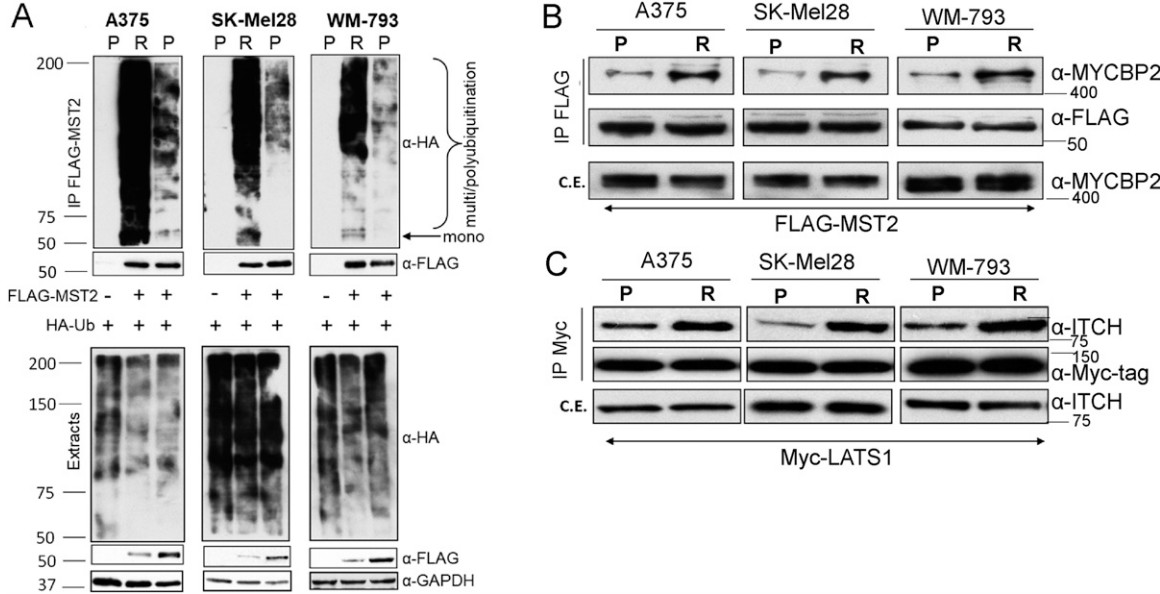

**Figure 4. MST2 and LATS1 exhibit increased ubiquitination and association with E3-ubiqitin ligases in resistant melanoma cells.**
**(A)** Parental and resistant A375, SK-Mel28, and WM-793 cell lines were transfected with HA-ubiquitin (1 $\mu g/2 \times 10^6$) alone or together with FLAG-MST2 (0.5 $\mu g/2 \times 10^6$) and lysed 48 h after transfection. Cell extracts were immunoprecipitated with FLAG antibody beads and ubiquitination of precipitated MST2 was determined using HA antibody. Expression of proteins was determined in cells extract by immunoblotting with specific antibodies. **(B)** Parental and resistant cells were transfected with FLAG-MST2 (0.5 $\mu g/2 \times 10^6$) and exogenous protein was IP 48 h after transfection using FLAG beads. Interaction with endogenous MCYBP2 was determined by immunoblotting with specific antibody. c.e., cell extracts. **(C)** Parental and resistant cells were transfected with Myc-LATS1 (1 $\mu g/2 \times 10^6$) and cell lysates were incubated with Myc-tag antibody to immunoprecipitate Myc-LATS1. ITCH interaction with Myc-LATS1 was determined using a specific antibody. In all blots numbers indicate molecular weight (kD). Source data are available for this figure.

LATS1 in the PLX4032 resistant cells, we co-transfected the cells with HA-tagged ubiquitin and FLAG-tagged MST2 and determined changes of ubiquitination by immunoprecipitating FLAG-MST2. Despite similar ubiquitination levels in the total cell lysates, MST2 ubiquitination was clearly enhanced in the resistant cell lines and correlated with lower expression levels of FLAG-MST2 in the cell extracts (Fig 4A). Similarly, expression of Myc-tagged LATS1 in WM-793 and SK-Mel28 cells showed increased ubiquitination in the resistant variants (Fig S4A). These experiments suggest that the expression of MST2 and LATS1 in PLX4032 resistant cells is down-regulated by ubiquitin-dependent proteasomal degradation.

To identify the E3 protein ligases responsible for this ubiquitination, we mined two interaction proteomic datasets produced in our group that have been previously used to map interactors of these kinases (Novacek et al, 2020; Quinn et al, 2021). In a FLAG-MST2 AP-MS experiment, we identified MYCBP2 as a binding interactor of MST2. MYCBP2 is an atypical E3 ubiquitin-ligase that catalyses the ubiquitination of threonine and serine rather than lysine residues (Pao et al, 2018). Interestingly, when validating the MST2-MYCBP2 interaction by co-immunoprecipitation–Western blotting, we not only observed the interaction in the parental cell lines, but also saw a clear increase of this interaction in the resistant cells (Figs 4B and S4B). Similarly, we identified the E3 ubiquitin ligase ITCH as a LATS1 interactor in Myc-LATS1 IPs (Figs 4C and S4B). This interaction was also enhanced in resistant cells. These data indicate that these E3-ligases might be responsible for the increase of MST2 and LATS1 ubiquitination in the resistant cells. The mechanism seems to be enhanced binding, as the total expression

levels of MYCBP2 and ITCH were unchanged between parental and resistant cells.

**Proteasome inhibition promotes cell death in MM resistant cell lines though and MST-dependent mechanism**

The previous data suggested that ubiquitination and proteasomal degradation of MST2 pathway proteins play an important role in the acquisition of resistance to BRAFi. This was further supported by the observation that the proteasome inhibitor bortezomib selectively induced PARP cleavage (a marker of apoptosis) in BRAFi resistant A375-R and SK-Mel28-R cells but not in parental cells (Figs 5A and S3). In WM-793 cells, PARP was overexpressed in resistant cells and showed constitutive cleavage which was increased by bortezomib. Based on this observation, we hypothesised that the enhanced degradation of MST2 pathway proteins prevents BRAFi induction of apoptosis in resistant cells, and that bortezomib can restore apoptosis by stabilizing MST2 pathway proteins. To test this, we treated the resistant cell lines with increasing amounts of bortezomib. We observed that bortezomib caused a clear dose dependent increase in MST2 protein abundance in all three resistant cell lines (Fig 5B). This was accompanied by an increase in cell death when the cells were treated with more than 30 nM of bortezomib (Fig 5C). Expression of specific MST2 siRNA significantly reduced bortezomib-induced cell death, further confirming the role of the MST2 pathway in mediating the response to bortezomib treatment. Of note, we also tested whether bortezomib can re-sensitize resistant cells to PLX4032, but the combination was toxic with a very

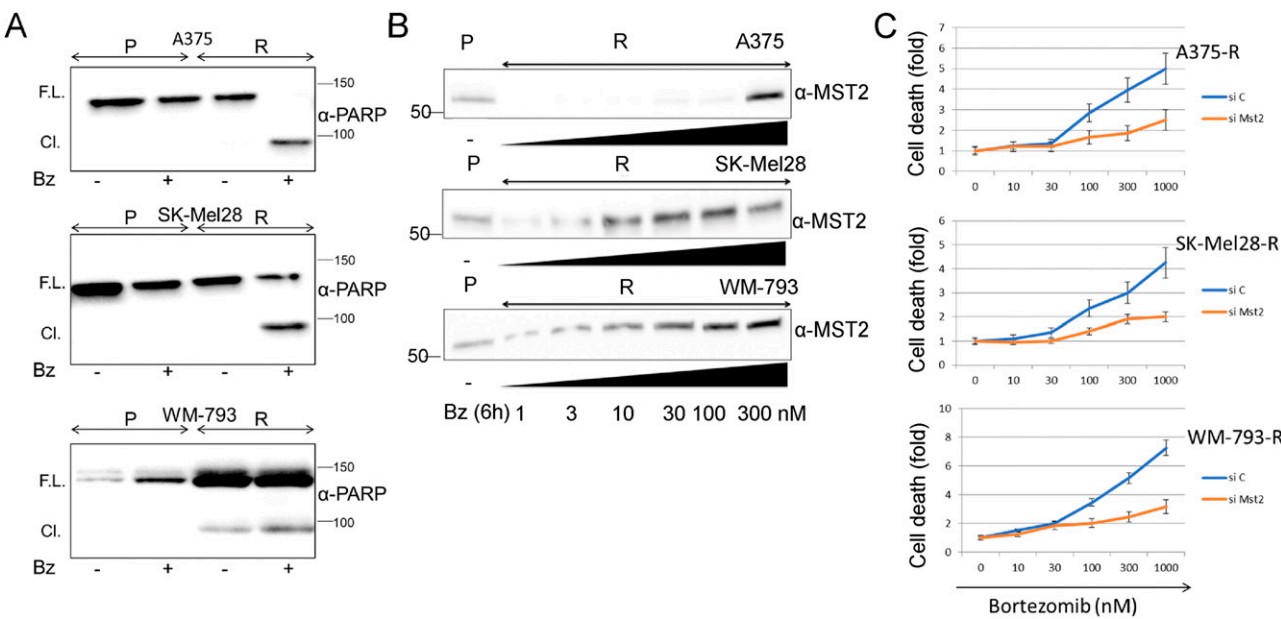

**Figure 5. Bortezomib induce apoptosis in PLX4032 resistant cells in an MST2 dependent fashion.**
**(A)** Parental and resistant A375, SK-Mel28, and WM-793 were treated with bortezomib (10 nM) for 6 h and cell lysates were immunoblotted with PARP antibody to determine caspase expression and cleavage. **(B)** A375, SK-Mel28, and WM-793 resistant cell lines were treated with the indicated concentrations of bortezomib and MST2 expression levels were monitored by immunoblotting with specific antibody. **(C)** A375, SK-Mel28, and WM-793 resistant cell lines were transfected with non-targeted (si C) or MST2 siRNA (siMST2) and after 24 h of transfection the cells were treated with the indicated concentrations of bortezomib for 3 h and cell death was determined by assaying DNA fragmentation by FACS. Error bars show SD, n = 3. In all blots numbers indicate molecular weight (kD).
Source data are available for this figure.

small therapeutic window, making a clear assessment of possible re-sensitisation or synergy difficult.

### Loss of expression of MST2 may correlate with development of resistance to BRAFi in patients

The data described above indicate that the reduction of MST2 pathway protein expression contributes to the development of resistance to vemurafenib. As a first attempt to test if this might be relevant in the clinical setting, we used a small cohort of nine human metastatic melanoma patients who were treated with vemurafenib alone or in combination with MEK inhibitors, and from whom pre-treatment and relapse histological samples were available from previous studies (Frederick et al, 2013) (Figs 6, S5, and S6 and Table 1). Although MST2 expression was detectable in melanomas before treatment, it seems to be lost or reduced in eight out the nine patients, who relapsed or progressed on vemurafenib treatment. Although the sample size is small, these data suggest that a reduction in MST2 expression is part of the mechanism how melanomas develop resistance to BRAFi in patients.

## Discussion

Malignant cutaneous melanoma is hallmarked by a hyperactivation of the RAS-RAF-MEK-ERK pathway that drives several aspects of

malignant transformation, including proliferation, survival, invasiveness and metastatic spread. All three most common driver mutations (*BRAF*, *NRAS*, and *NF1* mutations) converge on activating ERK signalling (Conway et al, 2020). About 10 yr ago, the introduction of vemurafenib revolutionized the treatment of metastatic melanoma patients with *BRAF*^V600E mutations (Bollag et al, 2010). However, despite good response rates and remarkable efficacy, most patients became resistant within 6–8 mo. The addition of MEKi synergizes with BRAFi (Sturm et al, 2010) improving the clinical efficacy and delaying the onset of resistance by ca. 3 mo (Flaherty et al, 2012). Later, BRAFi generations suffered from the same problem, which triggered numerous studies analysing resistance mechanisms (Sanchez et al, 2018; Rossi et al, 2019; Czarnecka et al, 2020; Lee et al, 2020; Proietti et al, 2020; McKenna & García-Gutiérrez, 2021). Despite a variety of mechanisms were discovered, the large majority shares the re-activation of ERK signalling as common theme. However, another interesting observation was that the various ways to reactivate ERK rested mainly on network rewiring rather than mutations that prevented drug binding to the target. Importantly, our results show that the BRAFi resistant cell lines we have generated show a rewiring of signalling pathways that include cell specific changes in RTK signalling and in the case of A375 acquisition of secondary mutations. These changes are likely affecting feedback mechanism and different signalling modules other than ERK pathway. This finding prompted us to investigate the role of pathways that crosstalk with RAF.

Although phosphorylation and activation of MEK1/2 is the only widely accepted catalytic function of the RAF kinases, they can

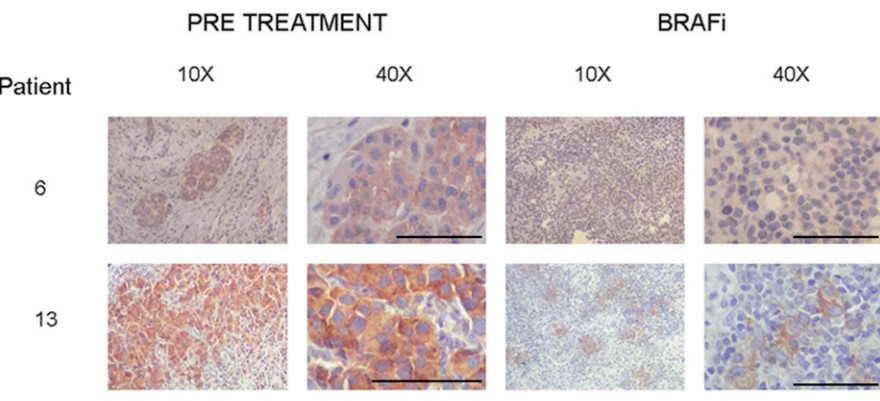

**Figure 6. MST2 expression is decreased in patients with metastatic melanoma treated with BRAF inhibitors.**
Immunohistochemistry (10× and 40× magnification) for the MST2 expression was detected as explained in materials and methods. Tumours biopsied from representative patients were stained with haematoxylin and eosin. The figure shows expression of MST2 in patients 6 and 13 before treatment (pre-treatment) and after treatment with BRAF (dabrafenib) and MEK inhibitors (BRAFi). Bar = 100 $\mu$m.

**Table 1. Loss of MST2 in eight of nine patients with relapsed or progressive disease.**

| Patient | Age | Sex | Treatment | Response | PFS | Censure_PFS | OS | Censure_OS | MST2 pre-treatment | MST2 relapse/progression |
|---------|-----|-----|-----------|----------|-----|-------------|-----|------------|---------------------|---------------------------|
| 6 | 74 | F | BRAFi + MEKi | PR (−59.9%) | 727 | 1 | 1,262 | 1 | ++++ | ++ |
| 9 | 50 | M | BRAFi + MEKi | PR (−45%) | 220 | 1 | 566 | 1 | ++ | − |
| 13 | 69 | M | BRAFi + MEKi | PR (−57.9%) | 275 | 1 | 344 | 1 | +++++ | ++ |
| 20 | 53 | F | BRAFi | PR (−51.2%) | 175 | 1 | 576 | 1 | +++ | ++++ |
| 24 | 68 | F | BRAFi | PR (−53%) | 114 | 1 | 335 | 1 | ++ | − |
| 25 | 72 | M | BRAFi + MEKi | PR (−64%) | 101 | 1 | 381 | 1 | ++ | − |
| 34 | 33 | M | BRAFi + MEKi | PR (−48.6%) | 483 | 1 | 640 | 1 | +++ | + |
| 35 | 64 | F | BRAFi + MEKi | PR (−38.6%) | 279 | 1 | 3,304 | 0 | ++++ | − |
| 42 | 70 | M | BRAFi + MEKi | PR (−76.1%) | 386 | 1 | 738 | 1 | +++ | + |

Histopathological slides of tumour samples taken before treatment and upon disease relapse or progression were stained for MST2 expression. MST2 expression is indicated by "+" on a scale from 1 to 5 (weak–very strong); "−" means no MST2 expression detectable. PFS, progression free survival; OS, overall survival. PFS and OS indicate days relative to treatment start date. Censure_PFS: 1 = progressed, 0 = not progressed. Censure_OS = dead, 0 = alive. PR, partial response. BRAFi single treatment with vemurafenib. Combination MEKi treatment with trametinib and BRAFi dabrafenib.

regulate other signalling pathways independent of their kinase activity (Baccarini, 2005; Rauch et al, 2011; Nolan et al, 2021). These include the suppression of proapoptotic pathways mediated by ASK1 (Chen et al, 2001) or MST2 (O'Neill et al, 2004). The connection with the MST2 pathway seemed particularly interesting, because several components of the MST2 pathway have been previously implicated in melanoma albeit in different roles. Overexpression of YAP1, a main transcriptional effector of MST2 signalling, can drive melanomagenesis (Thompson, 2020). By contrast, RASSF1A, an upstream activator of MST2, and LATS1, a downstream substrate of MST2, may work as tumour suppressors. RASSF1A expression is commonly lost in melanoma patients (Reifenberger et al, 2004), and the expression of LATS1 is often down-regulated (Yuan et al, 2015; Han et al, 2021). These observations are consistent with a tumour suppressive function of MST2 signalling that is independent of YAP1. Interestingly, we have previously identified such a pathway where MST2-LATS1 signalling leads to the stabilization of the TP53 tumour suppressor protein and subsequent apoptosis in a YAP1 independent fashion (Matallanas et al, 2011b). TP53 is mutated in 19% of melanoma patients in the TCGA database, and these mutations seem to be mutually exclusive with MST1/2 amplifications occurring in 7% of melanoma patients. Although more data are necessary to

get statistical confirmation for this inverse correlation, this potentially intriguing relationship warrants a study in its own right.

In the present study, we have focussed on the crosstalk between the RAF and MST2 pathways. In our original study (O'Neill et al, 2004) we found that RAF1 could bind to and inhibit MST2, whereas BRAF was rather inefficient in binding MST2. Here, we confirm the weak binding of wild-type BRAF to MST2 but find that BRAF$^{V600E}$ binds strongly to MST2. Vemurafenib treatment releases this inhibitory interaction resulting in MST2 activation and apoptosis that at least in part is MST2 dependent. Our results show that this interaction is also disrupted by other BRAFi, including inhibitors with different mechanisms of action indicating that MST2 activation might be a common mechanism of action for these therapeutics. An inhibitory interaction between BRAF$^{V600E}$ and the related MST1 kinase was observed previously in thyroid cancer (Lee et al, 2011). These results suggest that BRAF$^{V600E}$ has an altered conformation that not only enhances its kinase activity towards MEK but also its ability to inhibit MST2 proapoptotic signalling. This may thwart the switch-like relationship between the RAF1 and MST2 pathways, where RAF1 can either activate MEK to drive proliferation or inhibit MST2 to block apoptosis (Romano et al, 2014b). To phosphorylate MEK, RAF1 must release MST2, thereby coupling MEK driven proliferation to a

higher risk of apoptosis-mediated by MST2. As PLX4032 releases MST2 from BRAF$^{V600E}$ leading to apoptosis, it is very plausible that the acquisition of drug resistance not only requires re-activation of ERK signalling but also neutralization of the proapoptotic signals mediated by the MST2 pathways. Our results suggest that PLX4032 resistant cells can solve this dilemma by down-regulating the expression of MST2 and LATS1 proteins. Our data show that this down-regulation seems to be independent of ERK pathway reac-tivation. This seems not only to be happening in cell lines but also in melanoma patients emphasizing the urgency to identify a mech-anism and potential target for therapeutic interference.

Our results suggest that MST2 and LATS1 protein expression in resistant cells is down-regulated because of enhanced ubiq-uitination and subsequent proteasomal degradation. Using AP-MS we also identified the ubiquitin ligases as MYCBP2 and ITCH, re-spectively. MYCBP2 is an unconventional E3 ubiquitin ligase that modifies threonine and serine instead of lysine residues (Pao et al, 2018). It has few known substrates, suggesting a high substrate specificity, and an unusual catalytic mechanism (Pao et al, 2018), which may make it an attractive drug target. Intriguingly, MYCBP2 is highly expressed in skin (Uhlén et al, 2015), and the phosphorylation of the major MYCBP2 phosphorylation site, serine 3505, is down-regulated by both PLX4032 as well as the MEK inhibitor AZD6244 (selumetinib) (Stuart et al, 2015). Unfortunately, the functional consequence of this phosphorylation is unknown, but may suggest that the RAF pathway regulates MYCBP2 and thereby influences MST2 degradation. However, inhibiting the ERK pathway using RAFi and MEKi had no effect on the stability of MST2 pathway proteins suggesting that the reduction of MST2 pathway protein expression and activity is not a direct consequence of ERK pathway activity.

ITCH is a classic E3 ubiquitin ligase that can ubiquitinate RASSF1A (Pefani et al, 2016) and LATS1 (Ho et al, 2011; Salah et al, 2011) and mark them for degradation. The ITCH-mediated reduction of RASSF1A and/or LATS1 expression promotes cell proliferation, epithelial–mesenchymal transition, and tumorigenicity. Vice versa, knocking down ITCH expression increased LATS1 expression resulting in reduced proliferation and increased apoptosis (Ho et al, 2011; Salah et al, 2011). Thus, ubiquitination plays an important role in regulating the biological activity of the MST2 pathway. Impor-tantly, the relevance of ubiquitin-mediated degradation of the kinases of the pathway is supported by a recent study showing that the E3 ubiquitin ligase RNF6 interacts with MST1 promoting the degradation of this kinase in breast cancer which is associated with short survival (Huang et al, 2022).

Interestingly, the total abundance of ITCH and MYCBP2 was not altered in resistant cells, suggesting that the enhanced association between them and MST2 and LATS1 may depend on affinity changes caused by post-translational modifications (PTMs). This will require mapping the exact binding sites and PTMs that map to these sites or can alter their conformation. These analyses may reveal new druggable targets that could be used to overcome BRAFi resistance in melanoma. Here, we have concentrated on restoring MST2 and LATS1 expression using proteasome inhibitors. Indeed, elevation of MST2 and LATS1 protein concentrations induced apoptosis in PLX4032 resistant melanoma cells. Combination with PLX4032 proved very toxic preventing us to determine whether these combinations are synergistic. This is in contrast to thyroid cancer,

where PLX4032 and bortezomib synergised to inhibit BRAF$^{V600E}$ transformation (Tsumagari et al, 2018). Thus, in melanoma rather than a restoration of sensitivity to PLX4032, proteasome inhibition and elevation of MST2 and LATS1 protein abundances seems sufficient for triggering apoptosis. Interestingly, pharmacological screens have pointed to proteasome inhibitors selectively being active against cancer cells expressing BRAF$^{V600E}$ (Zecchin et al, 2013). In summary, our findings show that the crosstalk between the RAF and MST2 pathways could provide a target that may overcome resistance to BRAFi.

# Materials and Methods

## Cell culture and transfection

A375, SK-Mel28, WM-793, and SK-Mel2 were obtained from ATCC and were validated by sequencing before initiation of the study. All cells were grown in RPMI media (Gibco) supplemented with 10% foetal bovine serum (Gibco) and 2 mM L-glutamine (Gibco). Media for the resistant cells contains 2 $\mu$M PLX4032 (Selleck chemicals). HeLa cells were also validated by sequencing and grown in DMEM (Gibco) 10% foetal bovine serum and 2 mM L-glutamine. The cells were main-tained at 37°C and 5% CO$_2$. Cells were transfected with Lipofect-amine 2000 (Invitrogen) following the manufacturer's protocol and the amount of DNA and siRNA is indicated in each experiment. FLAG-BRAF, -BRAF$^{R509H}$, -BRAF $^{V600E/R509H}$, and BRAF$^{V600E}$, MYC-LATS1, and FLAG-MST2 have been described (Romano et al, 2014a; Jambrina et al, 2016; Quinn et al, 2021) HA-ubiquitin was a gift from Edward Yeh (plasmid # 18712; Addgene) (Kamitani et al, 1997). MST2 SMARTpool siRNA is from Dharmacon (M-012200) and was validated before (Matallanas et al, 2007). Bortezomib (S1013), trametinib (S2673), dabrafenib (GSK2118436), encorafenib (LGX818), sorafenib tosylate (S1014), and TAK-632 are from Selleck Chemicals and Proteasome inhibitor I from Calbiochem (539160).

## Generation of resistant cell lines

A375, SK-Mel28, and WM-793 cells were grown in medium containing increasing amounts of PLX4032 for 6 mo. The cells were initially grown in 10 nM PLX4032-containing medium for 1 mo, and cells that proliferated in this condition were expanded after 2-wk subculture for three times. Subsequently, the cells were grown in 30 nM PLX4032 and the same sequence was followed (grow for 3 wk and subculture three times). Cells that resisted these conditions were further expanded following the same culture process in increasing amounts on PLX4032 following this sequence 100, 30, 300 mM, and 1 $\mu$M. Finally, after ~6 mo (28–30 passes), the cells were resistant to 3 $\mu$M PLX4032. Resistant clones were split into different plates, frozen, and stored in liquid nitrogen to create cell stocks.

## Cell lysis, immunoprecipitation, and Western blotting

Cellular extracts from the different cell lines were produced using cell lysis buffer containing 150 mM NaCl (Sigma-Aldrich), 20 mM Hepes (Sigma-Aldrich), pH 7.5, 1% NP-40 NP 40, and proteases

inhibitors. Immunoprecipitation was performed by adding primary antibody and 5 μl of agarose beads to cell extracts. 5 μl of FLAG beads (Sigma-Aldrich) were added for FLAG IPs. The mix was incubated rotating for 2 h at 4°C followed by washes with washing buffer (150 mM NaCl, 20 mM Hepes, pH 7.5, and 0.5% NP-40). Immunoprecipitates or total cell lysates were separated by SDS polyacrylamide gel electrophoresis, followed by transfer to polyvinylidene fluoride (PVDF) membranes for Western blotting. The following antibodies were used: FLAG-M2 (A8592), ERK1/2 (M5670), and phospho-ERK1/2 (M8159), BRAF$^{V600E}$ (SAB5600047) from Sigma-Aldrich; pan-Ras (op40) from Calbiochem; HA-HRP (3F10) from Roche; HA (sc-7392), MST2 (SC-6213), anti-LATS1 (sc-9388 and sc-12494), YAP1 (SC-15407), BRAF (Sc-5284), HRAS (SC-520), KRAS (SC-30), and NRAS (SC519) from Santa Cruz; anti-RASSF1A (14-6888-82) from eBioscience; STK3 (MST2, ab52641), and MYCBP2 (Ab86078) from Abcam; phospho-MST1/2 (3682), phospho-LATS1 (8654), GAPDH (2118), vinculin (4650), and Myc-TAG (2276) from Cell Signaling; C-Raf (610152) PARP (556362) and ITCH (611199) from BD transduction laboratories; and ARAF (R14320) from Transduction Lab.

## Cell death and apoptosis assays

Cell death was analysed by flow cytometry as descried before (O'Neill et al, 2004). Briefly, cells were grown as indicated, and the medium containing floating cells was collected before trypsinization of plates. Trypsinized cells and the collected medium were pooled, and cells were pelleted by centrifugation. The cells were washed by resuspension in PBS, and after centrifugation, they were fixed with 90% EtOH/PBS for 1 h. The cells were incubated with propidium iodide and RNAse dissolved in PBS, and the population containing fragmented sub-G1 DNA content was measured using an Accuri C6 Flow Cytometer. In addition, activation of apoptosis was monitored by assessing PARP cleavage using Western blots.

## RTK signalling antibody array

Lysates from parental and resistant cells were used to screen activation of receptor tyrosine kinases using the PathScan RTK Signaling Antibody Array Kit (7982; Cell Signaling) following the manufacturer's instructions. This array contains phospho-specific antibodies, and equal amounts of proteins were loaded on the arrays.

## RAS activity assays

Ras activation was determined by performing RAF-RAS binding domain (RBD, which binds activated Ras) pull-down assays as previously described (Herrero et al, 2020). Briefly, cells were serum deprived for 16 h and lysed using magnesium rich buffer (25 mM Hepes, pH 7.5, 10 mM MgCl₂, 150 mM NaCl, 0.5 mM EGTA, 20 mM β-glycerophosphate, 0.5% nonidet-P40, 10% glycerol, and phosphatase- and protease inhibitors). The lysates were incubated with beads carrying recombinant GST-RBD protein and rotated for 1 h at 4°C. After washing with lysis buffer, activated Ras pulled down by the GST-RBD beads was measured by Western blotting. Western blots were quantified using ImageJ and the level of Ras activation was determined by calculating the ratio of Ras identified in GST-RBD pull downs (GTP-RAS) divided by the amount of Ras detected in total cell extracts.

## RT-PCR

Parental and resistant cells were lysed and total RNA was extracted as previously published (Duffy et al, 2014). Briefly, an RNeasy kit (QIAGEN) was used to extract total RNA, and cDNA was produced using QuantiTect Reverse Transcription kit (QIAGEN) following the manufacturer's instructions. An ABI 7900HT Real Time PCR System (Applied Biosystems) and TaqMan reagents (Applied Biosystems) were used according to the manufacturer's protocol. Gene expression was normalized to the expression of β-actin (Assay ID: 4326315E) with P0 (RPLP0, Assay ID: 4310879E) as a second endogenous control. Gene assays used were MST1 (STK4, Hs00178979_m1*), MST2 (STK3, Hs00169491_m1*), LATS1 (Hs01125523_m1*), LATS2 (Hs00324396_m1*), and YAP1 (Hs00902712_g1*). Biological duplicates were generated for all samples at all time points. Technical replicates for every sample and time point were also performed.

## Patient samples and immunohistochemistry

Patients with metastatic melanoma containing BRAF$^{V600E}$ mutation (confirmed by genotyping) were enrolled on clinical trials for treatment with a vemurafenib (BRAF inhibitor) or combined dabrafenib (another BRAFi) and trametinib (MEK inhibitor) and information about the study is included in Frederick et al (2013). Patients were consented for tissue acquisition per Institutional Review Board (IRB)-approved protocol (DFCI 11-181). Tumour biopsies were conducted pre-treatment (day 0), at 10–14 d on treatment and/or at time of progression. Formalin-fixed tissue was analysed to confirm that viable tumour was present via haematoxylin and eosin (H&E) staining. Responses were determined according to RECIST, version 1.1.

Patient tumours were fixed in 10% neutral buffered formalin as previously described (Frederick et al, 2013). Briefly, tissue was embedded in paraffin, and sectioned at five microns. Deparaffinized and rehydrated sections were subjected to epitope retrieval in 10 mM Tris–EDTA buffer, pH 9.0, and blocking in 3% BSA in TBST (Tris, pH 7.6, 0.05% Tween-20). Sections were incubated with MST2 (STK3) antibody (clone EP1466Y, Cat. no. 1943-1; Epitomics) for 1 h at RT. After peroxidase block in 3% H₂O₂, HRP-labelled anti-rabbit secondary antibody (Dako EnVision, K4003, RTU) was applied for 30 min. Slides were developed with DAB+ (K3468; Dako) and counterstained with haematoxylin (H-3401; Vector) before dehydration and mounting. Stained slides were scored by two experts, blinded pathologists for positivity and intensity.

## Statistics

The experiments were repeated at least three times unless indicated. Graphs were generated using Excel; error bars show SD.

# Data Availability

This study includes no new data deposited in external repositories. Proteomics data used in this study have been published before

(Novacek et al, 2020; Quinn et al, 2021) and are available in PRIDE repository, access numbers PXD018903 and PXD018905.

## Supplementary Information

## Acknowledgements

We thank the patients that donated the tumour samples used in this study. We thank Claudia Aura González (UCD, Conway Institute) for helping to score IHC samples. This work was supported by Science Foundation Ireland under grant numbers 18/SPP/3522 and 14/IA/2395 awarded to W Kolch, and CDA 15_CDA_3495 to D Matallanas. N Aboud is funded by UCD's School of Medicine Ad Astra Fellowship.

### Author Contributions

D Romano: conceptualization, data curation, formal analysis, investigation, and writing—review and editing.

L García-Gutiérrez: data curation, investigation, and writing—review and editing.

N Aboud: investigation.

DJ Duffy: data curation, investigation, and writing—review and editing.

KT Flaherty: data curation, investigation, and writing—review and editing.

DT Frederick: data curation, investigation, and writing—review and editing.

W Kolch: conceptualization, formal analysis, supervision, funding acquisition, investigation, and writing—original draft, review, and editing.

D Matallanas: conceptualization, data curation, formal analysis, supervision, funding acquisition, investigation, and writing—original draft, review, and editing.

### Conflict of Interest Statement

The authors declare that they have no conflict of interest.

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
