## [Reviewer comments · Life Science Alliance]

Life Science Alliance

Proteasomal downregulation of the MST2 pathway contributes to BRAF inhibitor resistance in melanoma

David Romano, Lucia Garcia-Gutierrez, Nourhan Aboud, David Duffy, Keith Flaherty, Dennie Frederick, Walter Kolch, and David Matallanas

DOI: <https://doi.org/10.26508/lsa.202201445>

Corresponding author(s): David Matallanas, University College Dublin and Walter Kolch, University College Dublin

Review Timeline:

Submission Date:	2022-03-11
Editorial Decision:	2022-04-21
Revision Received:	2022-07-05
Editorial Decision:	2022-08-01
Revision Received:	2022-08-15
Accepted:	2022-08-19

Transaction Report:

April 21, 2022

Re: Life Science Alliance manuscript #LSA-2022-01445-T

Dr. David Matallanas
University College Dublin
Systems Biology Ireland
Belfield
Dublin 4
Dublin, Dublin 4 D4
Ireland

Dear Dr. Matallanas,

Thank you for submitting your manuscript entitled "Proteasomal downregulation of the MST2 pathway contributes to BRAF inhibitor resistance in melanoma" to Life Science Alliance. The manuscript was assessed by expert reviewers, whose comments are appended to this letter. We invite you to submit a revised manuscript addressing the Reviewer comments.

Thank you for this interesting contribution to Life Science Alliance. We are looking forward to receiving your revised manuscript.

Sincerely,

B. MANUSCRIPT ORGANIZATION AND FORMATTING:

Reviewer #1 (Comments to the Authors (Required)):

Romano et al. describe a novel mechanism of BRAF inhibitor resistance which relies on the downregulation of pro-apoptotic MST2 pathway components, such as MST2 itself, LATS1 and YAP1. The authors nicely show that BRAFV600E mutant cells inhibit pro-apoptotic MST2 signaling, by a direct interaction of the mutant BRAFV600E protein with the MST2 kinase. This inhibitory interaction is a known kinase independent function of wildtype RAF1 to block this pro-apoptotic pathway and thereby promoting survival, but hasn't been analyzed in the context of BRAF inhibitor resistance so far. In BRAF inhibitor sensitive melanoma cells, this inhibitory interaction is rapidly lost after drug treatment and MST2 signaling is induced. Importantly, BRAF inhibitor resistant cells, that fail to induce apoptosis in response to the inhibitors, show a downregulation of several components of the MST2 pathway including MST2, LAST1, RASSF1A and YAP1. Mechanistically, the authors show that this is not due to a transcriptional down-regulation, but rather caused by increased ubiquitination and degradation of the MST2 pathway components. The E3 ligases MYCBP2 and ITCH, that are identified as interaction partners of MST2 and LATS1, respectively, could be responsible for the ubiquitination and subsequent degradation. Therapeutically interesting is their observation that proteasome inhibitors are able to induce cell death in the resistant cells. Analysis of a small patient cohort confirmed the downregulation of the MST2 protein in patients that progressed or relapsed after treatment with BRAF inhibitors or a combination of BRAF and MEK inhibitors. The paper by Romano and colleagues offers an advantage to the field as it describes a novel resistance mechanism to BRAF inhibitors.

All main points of the paper are convincingly supported by experimental data. Some changes in the text and additional experiments could help to improve the message of the paper and will be addressed in the following. This includes some additional Western blot and Co-immunoprecipitation experiments and should be achievable in a short amount of time (2-3 month of revision).

1)The authors nicely show that MST2 strongly interacts with mutant BRAFV600E in cells overexpressing this oncogene, but also in melanoma cells that endogenously express the mutant BRAF protein. Inhibition of BRAF signaling in these cells leads to loss of this inhibitory interaction, MST2 pathway activation and eventually to reduced cell viability. Resistant cells evade this route of cell death induction by downregulating the major components of this pathway. This is nicely shown in Fig. 2C. Although MST2 protein amount is strongly reduced, there is a significant amount of protein left. It would be interesting to analyze by immunoprecipitation experiments if the inhibitory interaction of BRAF and MST2 persists in the resistant cells. In particular, as phospho-MST2 levels are not upregulated at all after PLX4032 treatment in the resistant cells, one could hypothesize that the interaction persists despite the presence (and binding of PLX4032). In this case it would be interesting to discuss which pathways (AKT activation and phosphorylation of MST2, maintained ERK activity?) are responsible for the drug induced dissociation or re-association between MST2 and BRAFV600E.

In this regard, it would also be interesting to test if other BRAF inhibitors like Encorafenib, Dabrafenib or a dimer-breaking inhibitor such as PLX8394 induce a similar release of MST2 from BRAFV600E.

2)Combined BRAFi and MEK inhibition is able to reduce MEK/ERK signaling of the resistant cells and most likely also their viability. It would be interesting to analyze the expression and activation status of the MST2 pathway in the combinatorial treated cells as well. This would clarify how reversible the Ubiquitination-dependent downregulation of the MST2 pathway components is and if it is dependent on ERK activity.

3)The immunoprecipitation data shown in Fig. 4B/C could be improved by including empty vector controls. The first experiment demonstrates enhanced binding of the E3 ligase to Flag-MST2 in resistant compared to parental cells. Including cells without Flag-MST2 expression (expressing an empty vector control) would allow to determine if the MYCBP2 signal in the parental cells is due to unspecific binding of MYCBP2 to the beads, inefficient washing or indeed is a specific signal indicating association between the two proteins. The same holds true for the IP shown in Figure 4C. Including this control would strengthen the data. Nevertheless, the clear increase in resistant cells is convincingly shown by the data and the conclusions that the authors draw from the data are valid.

4) How do parental cancer cells react to Bortezomib treatment? Fig. 3B only shows the increased protein abundance in resistant cells after treatment. Wouldn't treatment of the parental cells most likely also affect the physiological turnover rate of e.g. MST2 (as it is also ubiquitinated, see Fig. 4A)? It would be helpful to compare the relative increase in protein abundance between parental and resistant cells to draw the conclusion that MST2 proteins are particularly affected by degradation in resistant cells. Similarly, although Fig. 5A shows nicely that Bortezomib does not induce PARP cleavage at a concentration of 10 nM in parental cells, cell death of the resistant cells, however, only starts with 30 nM (Fig. 5C and main text p.7). Is cell death induction induced by Bortezomib specific to resistant cells also at 30 nM and higher concentrations?

5) The patient data provided in Figure 6, Table 1, S4 and S5 are poorly described in regard to which drug combinations were used. In the methods part the authors write: "treatment with a Vemurafenib (BRAF inhibitor) or combined Vemurafenib and dabrafenib + trametinib (MEK inhibitor)", p.10/11. This reads like vemurafenib and dabrafenib were administered together with trametinib in a triple combination, which was not the case (see Frederick et al.) as patients were treated with either BRAF inhibitor alone (vemurafenib) or BRAF + MEK inhibition (dabrafenib + trametinib). Please clearly indicate which drug combinations were used and re-phrase main text ("patients who were resistant to vemurafenib treatment alone or in combination with MEK inhibitors", p.7), methods part (p.10/11) and figure legends of Fig. 6, S4, S5 and Table 1 accordingly. It would also help to re-organize the Supplementary Figure S4 and S5 by putting the "BRAFi only" and the "BRAFi+MEKi" patient samples into separate groups.

Additional issues (minor comments)

1) SK-Mel28 is mistakenly named SK-Mel20 (p.49)

2) S2A/B: The authors state in the main text of the results part (p.4) that all the resistant cells showed similar responses in terms of ERK re-activation and resistance to apoptosis, which is why they chose one representative clone from each line. The data for the second statement (apoptosis data for each clone) are not included in the supplementary Figure S2A/B. Please include these data or re-phrase.

3) Wrong figure link: PLX4032 combined with MEK inhibition caused ERK inhibition (Fig. S2D). It is Fig. 2D.

4) The labeling of the Western blot in Fig. 2E is incorrect. The bottom Ras Western blot should be labelled anti-HRas not anti-KRas.

5) Fig. 2F. The PathScan measures activation of RTK signaling by detection of phosphorylated tyrosine or other residues on the different targets. In the result description the authors do not clearly describe if expression or activation of the receptors is increased ("A375R showed an increase of PDGF-R and ALK" and "...increase of IGF-1R and the SRC effector SRC", p.5). It would be helpful to re-phrase this.

6) Please correct the following sentence: "Finally, WM793R had an increase of IGF-1R and the SRC effector SRC", p.5.

7) Fig. 4A: Labelling of +/- legend indicating where HA-Ubiquitin and Flag-MST2 is expressed is missing

8) Figure S3: The figure legend states that data of A375, Sk-Mel28 and WM-793 are shown, data on A375, however, are not shown in this figure. In addition, the title of the supplementary figure states that interaction data with E3 ligases are shown, these data, however, can be found in the main Figure 4 B/C. This needs to be corrected. Furthermore, the information on the amount of DNA used for transfection, should be given in relation to the number of cells or plate size.

9) Mistake in the results part describing Fig. 4C: "Similarly, we identified the E3 ubiquitin ligase ITCH as a LATS1 interactor in FLAG-LATS1 IPs (Fig. 4C), p.5". According to the figure and figure legend a MYC-LATS1 IP was performed. Please correct this mistake.

10) In general, the manuscript needs some more proof-reading: Cell line spelling is not consistent (WM-793 vs. WM793 etc.), abbreviations are not introduced (e.g. for the resistant cell lines or c.e. for cell extract), word duplications can be found and there are some typos (figure legend Fig. 3: PLX932 instead of PX4032)

Reviewer #2 (Comments to the Authors (Required)):

Romero D. and al present interesting data about the mechanisms of resistance to BRAFv600 inhibitors in melanoma carcinoma. They concentrate their study on the links between the acquisition of resistance and the regulation of MST2 kinase, an important tumour suppressor gene involved in the Hippo pathway. They first demonstrate in melanoma cell lines that MST2 activity is buffered through BRAFv600 interaction, leading to MST2 reactivation in presence of inhibitors of BRAFv600. Then, by comparing parental cell lines and their resistant counterparts engineered in vitro, they highlight that acquired resistance to BRAFv600 inhibitors is correlated to the destabilization of MST2 through an ubiquitin-proteasome dependent mechanism. Potential E3-ubiquitin ligases involved in this process are proposed. Finally, the authors present data demonstrating the ability of proteasome inhibitors to stabilize MST2 in resistant cells leading to their death by apoptosis. Clinical relevance of these data is also highlighted by showing in melanoma samples that MST2 is indeed downregulated when tumours acquire resistance. Overall, the data are rather convincing and I do not have major concerns and criticism about them. I would however recommend

the authors to complete and improve three main aspects of their data and to perform a couple of minor revision.

Main points:

1/ Outside of MST2 regulations, the authors highlight a couple of molecular regulations fostering ERK reactivation in the resistant cell lines. As indicated by the authors, it is well known that secondary mutation in other oncogenes such as NRAS can reactivate the MAPK pathway. Authors should analyze these mutations in their resistant cell lines, at least in the three RAS isoforms to really confirm that ERK reactivation is, in their experimental conditions, rather due to RTK reactivation. In line with these comments, authors have only highlighted the stronger activation of RTK signaling but have not commented on the presence of downregulations (some are obvious on the dot blot Fig. 2F). Finally, it would be interesting to discuss how could be related the mechanisms leading to ERK reactivation and to MST2 downregulation in resistant cells.

2/ Another conclusion requiring revisions concern the effects of proteasome inhibitors on MST2 stability and their ability to induce cell death. The authors do not present data about the inhibition of the proteasome in parental cell lines. Would MST2 stability also be regulated in a same manner? Authors show that proteasome inhibitors do not lead to apoptosis induction in parental cells. It would be therefore interesting to know if this data is linked to an absence of MST2 stabilization in this context. Otherwise, it would mean that an MST2 target is modified in R cells and would lay upstream of the apoptotic mechanisms. The authors should perform these experiments and comment these results.

3/ IHC data are not all convincing regarding the analyzed sections. The difference of expression of MST2 for the patient 9, 20, 34, 42 are therefore not obvious (supplementary figure 4 and 5). The authors should try to improve this. Also, to be more convincing, they may try to analyze other components of the MST2 pathway by analogy to their analyses performed in vitro (LATS1 and YAP for example)

Minor revisions :

1/ "These data indicated that the reduction in MST2, LATS1, and YAP1 proteins expression in PLX4032 resistant cells is due to the acquisition of resistance to the BRAF inhibitors the core proteins of the Hippo pathway are targeted for proteasomal degradation"

This sentence should be reviewed for clarity.

2/ Fig4A : - and + are not correctly labelled

Reviewer #3 (Comments to the Authors (Required)):

David Romano et al. studied the role of pro-apoptotic MST2 pathway in the acquisition of resistance to BRAFi in melanoma cells. They elegantly show that BRAFi leads to MST2 activation with concomitant increase in cell death. Importantly, this does not occur in in-house resistant cells as these proteins show lower levels of MST2, LATS1 and YAP. This is a consequence of increased ubiquitination, mediated at least in part by MYCBP2 and ITCH, that leads to increased proteasome degradation. Importantly, proteasome inhibition led to increased MST2-dependent cell death in resistant cells. Finally, the authors found MST2 downregulation in biopsies from patient that had progressed to BAFi Vemurafenib, alone or in combination with MEKi. Overall, the paper is very easy to read, very well written and structured, the experiments performed are adequate and the results, convincing. I've noted some minor text changes to be done and a few issues that I believe can be solved quite easily.

The MST2 pathway is inhibited by mutant BRAF in melanoma cells and the effect is rescued by BRAF inhibitors.

The data presented in figure 1 is well structured and convincing. When citing Fig. 1B, it says "Fig. 1b"

Acquisition of resistance to BRAF inhibitors causes the downregulation of MST2 pathway proteins.

The data presented in figure 1 is well structured and convincing. However, the authors claim that there's synergy between BRAFi and MEKi in the resistant cells in "Fig. S2C" when it should be "Fig. 2D". Looking at the WB, Trametinib alone prevents ERK phosphorylation (except in WM-793 cells, where there's a faint pERK band under Trametinib treatment) so it's impossible to assess whether there's synergy with PLX4032. Please, correct this and/or rephrase it. Also, in Fig. 2F and S2D they show the results from PathScan RTK signaling antibody array. Although the results are clear, the authors don't comment on other possible RTK signalling components that seem to be modulated in resistant cells. For instance (according to the PathScan datasheet): there's increased pSTAT1 in A375R whereas in this cell line IRS1 and Src (both activated in SkMel28R and Src being activated in WM-793) seem to be decreased. Also, S6 Ribosomal Protein shows modulation in the three cell lines though differently: decreased in A375R and SkMel28R but activated in WM-793. Among others, A375 has mutations in CDKN2A and VAV1, SkMel28 in Cdk4, PTEN, EGFR and p53 while WM973 in Cdk4 and PTEN. Maybe their mutational profiles could impact on the signalling rewiring in resistant cells. Please expand on this. Also, SkMel28 is incorrectly referred as SkMel20 the first time it appears. Please correct the following: "...Therefore, we assessed the protein expression of LAST1, RASSF1A", it's LATS1 and "...Finally, WM793R had an increase of IGF-1R and the SRC effector SRC...". SRC effector SRC.

MST2 pathway proteins are downregulated by proteasome degradation in BRAFi resistant cells.

The data presented in figure 3 is well structured and convincing. There is one sentence, though, that it's not clear, at least to me:

"These data indicated that the reduction in MST2, LATS1, and YAP1 proteins expression in PLX4032 resistant cells is due to the acquisition of resistance to the BRAF inhibitors the core proteins of the Hippo pathway are targeted for proteasomal degradation." Maybe "after acquisition of resistance to BRAFi, the core proteins of the Hippo pathway are targeted for proteasomal degradation, leading to lower protein levels"

Increased ubiquitination promotes MST2 and LATS1 proteasomal degradation in BRAFi resistant cell lines. The data presented in figure 4 well structured and convincing. In Fig. 4A, the indications for HA and FLAG are missing. The author state: "...these E3-ligases might be responsible for the decrease of MST2 and LATS1 ubiquitination in resistant cells". I believe that siRNA-mediated knock-down of MYCBP2 and/or ITCH could confirm this and add value to this finding. I would suggest the authors, if the Editor seems appropriate, to perform these experiments.

Proteasome inhibition promotes cell death in MM resistant cell lines though and MST-dependent mechanism. The data presented in figure 5 well structured and convincing. In figure legend: "Parental and resistant A375, SK-MEL28 and WM-793 were treater with". Please, correct "treater" into "treated".

Loss of expression of MST2 may correlate with development of resistance to BRAFi in patients
The data presented in figure 6 well structured and convincing. "...we used a small cohort of nine human metastatic melanoma patients who were resistant to vemurafenib treatment alone or in combination with MEK inhibitors, and where both pre-treatment...". Please, correct this "and where both". Maybe "from whom"

Discussion

In the third paragraph, the authors state: "the phosphorylation of the major MYCBP3 phosphorylation site, serine 3503, is downregulated by both PLX4032 as well as the MEK inhibitor AZD6244 (selumetinib) (Stuart et al., 2015)". I believe there's a mistake there since MYCBP3 doesn't exist and I couldn't find it in Stuart et al. paper. MYCBP2 phosphorylation sites don't include Ser 3503 but 3505. Please, correct and/or clarify this.

Reviewer #1 (Comments to the Authors (Required)):

Romano et al. describe a novel mechanism of BRAF inhibitor resistance which relies on the downregulation of pro-apoptotic MST2 pathway components, such as MST2 itself, LATS1 and YAP1. The authors nicely show that BRAFV600E mutant cells inhibit pro-apoptotic MST2 signaling, by a direct interaction of the mutant BRAFV600E protein with the MST2 kinase. This inhibitory interaction is a known kinase independent function of wildtype RAF1 to block this pro-apoptotic pathway and thereby promoting survival, but hasn't been analyzed in the context of BRAF inhibitor resistance so far. In BRAF inhibitor sensitive melanoma cells, this inhibitory interaction is rapidly lost after drug treatment and MST2 signaling is induced. Importantly, BRAF inhibitor resistant cells, that fail to induce apoptosis in response to the inhibitors, show a downregulation of several components of the MST2 pathway including MST2, LATS1, RASSF1A and YAP1. Mechanistically, the authors show that this is not due to a transcriptional down-regulation, but rather caused by increased ubiquitination and degradation of the MST2 pathway components. The E3 ligases MYCBP2 and ITCH, that are identified as interaction partners of MST2 and LATS1, respectively, could be responsible for the ubiquitination and subsequent degradation. Therapeutically interesting is their observation that proteasome inhibitors are able to induce cell death in the resistant cells. Analysis of a small patient cohort confirmed the downregulation of the MST2 protein in patients that progressed or relapsed after treatment with BRAF inhibitors or a combination of BRAF and MEK inhibitors. The paper by Romano and colleagues offers an advantage to the field as it describes a novel resistance mechanism to BRAF inhibitors.

All main points of the paper are convincingly supported by experimental data. Some changes in the text and additional experiments could help to improve the message of the paper and will be addressed in the following. This includes some additional Western blot and Co-immunoprecipitation experiments and should be achievable in a short amount of time (2-3 month of revision).

We thank the reviewer for the comments and have now completed several of the figures following these recommendation as explained below.

1)The authors nicely show that MST2 strongly interacts with mutant BRAFV600E in cells overexpressing this oncogene, but also in melanoma cells that endogenously express the mutant BRAF protein. Inhibition of BRAF signaling in these cells leads to loss of this inhibitory interaction, MST2 pathway activation and eventually to reduced cell viability. Resistant cells evade this route of cell death induction by downregulating the major components of this pathway. This is nicely shown in Fig. 2C. Although MST2 protein amount is strongly reduced, there is a significant amount of protein left. It would be interesting to analyze by immunoprecipitation experiments if the inhibitory interaction of BRAF and MST2 persists in the resistant cells. In particular, as 1hospho-MST2 levels are not upregulated at all after PLX4032 treatment in the resistant cells, one could hypothesis that the interaction persists despite the presence (and binding of PLX4032).

We had already tried this experiment several times and now include as Figure S2E. We could never see an interaction in those co-IP even when we increased the amount of material. This would indicate that the interaction is lost. However, we are still cautious since the levels of MST2 are very low in the resistant cells and we had to use a high number of cells to get these IPs and cell extracts. Thus, BRAF might still inhibit the remaining MST2 but the level of this protein in the co-IP could be below the level of detection of the WB. Now we mention this in the text "Additionally, we could not detect the interaction between BRAF and MST2 in A375R cells which may be due to the low level of expression of MST2 in these cells (Fig S2E)"

In this case it would be interesting to discuss which pathways (AKT activation and phosphorylation of MST2, maintained ERK activity?) are responsible for the drug induced dissociation or re-association between MST2 and BRAFV600E. In this regard, it would also be interesting to test if other BRAF inhibitors like Encorafenib, Dabrafenib or a dimer-breaking inhibitor such as PLX8394 induce a similar release of MST2 from BRAFV600E.

We have performed the experiment suggested by the reviewer and treated the cells with 3 BRAF inhibitors including type I½ and II BRAFi (vemurafenib is a type I½ inhibitor). The data indicate that these BRAFi have similar effects as vemurafenib, decreasing the MST2-BRAFV600E interaction. We have added a new figure showing this (Fig. S1A). The text now reads "To test if the disruption of this interaction is specific to this BRAFi we treated A375 cells with other BRAF inhibitors including 2 other type I½ and 2 type II inhibitors (Cook and Cook, 2021), and we could observe that all of them decrease MST2-BRAF interaction (Fig S1A)" and in the discussion "Our results show that this interaction is also disrupted by other BRAFi including with different mechanisms of action indicating that MST2 activation might be a common mechanism of action for these therapeutics."

2) Combined BRAFi and MEK inhibition is able to reduce MEK/ERK signaling of the resistant cells and most likely also their viability. It would be interesting to analyze the expression and activation status of the MST2 pathway in the combinatorial treated cells as well. This would clarify how reversible the Ubiquitination-dependent downregulation of the MST2 pathway components is and if it is dependent on ERK activity.

We have now included this experiment repeated in the A375 cells as supplementary figure S2D. We observe that both PLX4032 (RAFi) and trametinib (MEKi) slightly increase LATS1 activation in parental cells. Combining PLX4032 and trametinib does not further increase LATS1 activation. In the parental cells, we could not see any change of LATS1 in response to PLX4032 and trametinib alone or in combination, BRAFi indicating that the loss of LATS1 expression in resistant cells is not mediated by a loss of ERK signalling. In resistant cells the expression of LATS1 is reduced and its phosphorylation is not detectable. We now explain this in the text " Treatment of A375 cells with PLX4032 and trametinib alone or in combination resulted in a slight activation of LATS1 in parental but not in resistant cells. Importantly, none of the treatments had any impact on the expression or measurable activation of LATS1 in the resistant cells indicating that the reduction of the MST2-LATS1 pathway in resistant cells is not mediated by ERK1/2 (Fig. S2D).".

3) The immunoprecipitation data shown in Fig. 4B/C could be improved by including empty vector controls. The first experiment demonstrates enhanced binding of the E3 ligase to Flag-MST2 in resistant compared to parental cells. Including cells without Flag-MST2 expression (expressing an empty vector control) would allow to determine if the MYCBP2 signal in the parental cells is due to unspecific binding of MYCBP2 to the beads, inefficient washing or indeed is a specific signal indicating association between the two proteins. The same holds true for the IP shown in Figure 4C. Including this control would strengthen the data. Nevertheless, the clear increase in resistant cells is convincingly shown by the data and the conclusions that the authors draw from the data are valid.

We have now included this control in figure S4B. Of note, we had this control in the original IP that we used for the proteomics experiments which already showed specificity of the interaction. We always include these controls the first time we do an IP since we agree with the reviewer that it is a key control but for space and aesthetics reasons, we did not include it in every repeat of the experiment. The experiment included in the supplementary figure is one of the three repeats for A375 cells and clearly shows the specificity of this interaction. Reference to the figure is included in the text.

4) How do parental cancer cells react to Bortezomib treatment? Fig. 3B only shows the increased protein abundance in resistant cells after treatment. Wouldn't treatment of the parental cells most likely also affect the physiological turnover rate of e.g. MST2 (as it is also ubiquitinated, see Fig. 4A)? It would be helpful to compare the relative increase in protein abundance between parental and resistant cells to draw the conclusion that MST2 proteins are particularly affected by degradation in resistant cells.

This was actually an oversight on our side and we agree that this was an important experiment missing as also pointed out by reviewer 2. We have now performed this experiment and include it as Figure S3. Surprisingly, we observe that the inhibition of proteasome in the A375 and SK-Mel28 parental cells has very little effect in the level of expression of MST2 and LATS1. Therefore, it seems that the effect of proteasome in protein expression of the members of the pathway is increased during the acquisition of resistance to PLX4032 in these cells. This is now explained in the text "While in parental cells these inhibitors had little effect on the expression levels of the proteins of the MST2 pathway (Fig. S3), both inhibitors consistently and rapidly increased the expression of MST2, LATS1 and YAP1 in all three resistant cell lines (Fig. 3B)"

Similarly, although Fig. 5A shows nicely that Bortezomib does not induce PARP cleavage at a concentration of 10 nM in parental cells, cell death of the resistant cells, however, only starts with 30 nM (Fig. 5C and main text p.7). Is cell death induction induced by Bortezomib specific to resistant cells also at 30 nM and higher concentrations?

We blotted for PARP in Fig. S3 and we cannot see a major change of PARP cleavage in SK-MEL28 and A375 parental at higher concentrations therefore the effect of Bortezomib seems to be specific for the resistant cells. It must be noted that in A375 parental cells an increase of cell death in response to Bortezomib has been reported before ([10.1158/0008-5472.CAN-08-0426](https://pubmed.ncbi.nlm.nih.gov/10.1158/0008-5472.CAN-08-0426/)), but after 48 hours treatment.

5) The patient data provided in Figure 6, Table 1, S4 and S5 are poorly described in regard to which drug combinations were used. In the methods part the authors write: "treatment with a Vemurafenib (BRAF inhibitor) or combined Vemurafenib and dabrafenib + trametinib (MEK inhibitor)", p.10/11. This reads like vemurafenib and dabrafenib were administered together with trametinib in a triple combination, which was not the case (see Frederick et al.) as patients were treated with either BRAF inhibitor alone (vemurafenib) or BRAF + MEK inhibition (dabrafenib + trametinib). Please clearly indicate which drug combinations were used and re-phrase main text ("patients who were resistant to vemurafenib treatment alone or in combination with MEK inhibitors", p.7), methods part (p.10/11) and figure legends of Fig. 6, S4, S5 and Table 1 accordingly. It would also help to re-organize the Supplementary Fig. S4 and S5 by putting the "BRAFi only" and the "BRAFi+MEKi" patient samples into separate groups.

We apologise for the lack of clarity. We have now corrected the text to make clear the treatments done, and we have changed the labels and figure legends. The text now reads "As a first attempt to test if this might be relevant in the clinical setting, we used a small cohort of nine human metastatic melanoma patients who were treated with vemurafenib alone or in combination with MEK inhibitors, and from whom pre-treatment and relapse histological samples were available from previous studies". As suggested by the reviewer we have now separated the patients in two different groups and show them in separated figures.

Additional issues (minor comments)

- 1) SK-Mel28 is mistakenly named SK-Mel20 (p.49) Corrected
- 2) S2A/B: The authors state in the main text of the results part (p.4) that all the resistant cells showed similar responses in terms of ERK re-activation and resistance to apoptosis, which is why they chose one representative clone from each line. The data for the second statement (apoptosis data for each clone) are not included in the supplementary Figure S2A/B. Please include these data or re-phrase. We have rephrased as follows: "As all clones showed similar responses in terms of ERK re-activation and resistance to apoptosis, a representative clone from each resistant cell line is shown (Fig. 2A-E) and was used to perform most of the follow up experiments."
- 3) Wrong figure link: PLX4032 combined with MEK inhibition caused ERK inhibition (Fig. S2D). It is Fig. 2D. Corrected
- 4) The labeling of the Western blot in Fig. 2E is incorrect. The bottom Ras Western blot should be labelled anti-HRas not anti-KRas. Corrected
- 5) Fig. 2F. The PathScan measures activation of RTK signaling by detection of phosphorylated tyrosine or other residues on the different targets. In the result description the authors do not clearly describe if expression or activation of the receptors is increased ("A375R showed an increase of PDGF-R and ALK" and "...increase of IGF-1R and the SRC effector SRC", p.5). It would be helpful to re-phrase this.

We clarify this in the materials and methods section "This array contains phospho-specific antibodies and equal amounts of proteins were loaded on the arrays". The text has been also changed to indicate that there is a change in phosphorylation of the proteins monitored "A more likely explanation is that RAS signalling is caused by the activation of upstream regulators, which causes hyperactivation of several RAS isoforms. To test this, we used PathScan receptor tyrosine kinase (RTK) signalling antibody arrays. All cell lines had upregulated RTK signalling. A375R showed an increase phosphorylation of PDGF-R and ALK and a decrease of c-ABL and SRC phosphorylation. SK-MEL28 showed an increase IRS1 and SRC activation and lower ERK1/2 and S6RP phosphorylation (Fig. 2F and S2F). Finally, WM-793R had an increase of activation of IGF-1R, S6RP and the EGF-R effector SRC"
- 6) Please correct the following sentence: "Finally, WM793R had an increase of IGF-1R and the SRC effector SRC", p.5. Corrected
- 7) Fig. 4A: Labelling of + /- legend indicating where HA-Ubiquitin and Flag-MST2 is expressed is missing Corrected
- 8) Figure S3: The figure legend states that data of A375, Sk-Mel28 and WM-793 are shown, data on A375, however, are not shown in this figure. In addition, the title of the supplementary figure states that interaction data with E3 ligases are shown, these data, however, can be found in the main Figure 4 B/C. This needs to be corrected. Furthermore, the information on the amount of DNA used for transfection, should be given in relation to the number of cells or plate size. Corrected
- 9) Mistake in the results part describing Fig.4C: "Similarly, we identified the E3 ubiquitin ligase ITCH as a LATS1 interactor in FLAG-LATS1 IPs (Fig. 4C), p.5". According to the figure and figure legend a MYC-LATS1 IP was performed. Please correct this mistake. Corrected
- 10) In general, the manuscript needs some more proof-reading: Cell line spelling is not consistent (WM-793 vs. WM793 etc.), abbreviations are not introduced (e.g. for the resistant cell lines or c.e. for cell extract), word duplications can be found and there are some typos (figure legend Fig. 3: PLX932 instead of PX4032) We have corrected these typos.

Reviewer #2 (Comments to the Authors (Required)):

Romero D. and al present interesting data about the mechanisms of resistance to BRAFv600

inhibitors in melanoma carcinoma. They concentrate their study on the links between the acquisition of resistance and the regulation of MST2 kinase, an important tumour suppressor gene involved in the Hippo pathway. They first demonstrate in melanoma cell lines that MST2 activity is buffered through BRAFv600 interaction, leading to MST2 reactivation in presence of inhibitors of BRAFv600. Then, by comparing parental cell lines and their resistant counterparts engineered in vitro, they highlight that acquired resistance to BRAFv600 inhibitors is correlated to the destabilization of MST2 through an ubiquitin-proteasome dependent mechanism. Potential E3-ubiquitin ligases involved in this process are proposed. Finally, the authors present data demonstrating the ability of proteasome inhibitors to stabilize MST2 in resistant cells leading to their death by apoptosis. Clinical relevance of these data is also highlighted by showing in melanoma samples that MST2 is indeed downregulated when tumours acquire resistance. Overall, the data are rather convincing and I do not have major concerns and criticism about them. I would however recommend the authors to complete and improve three main aspects of their data and to perform a couple of minor revision.

We thank the reviewer for the suggestions to improve the manuscript. We have now addressed the points raised by the reviewer which in several cases were similar to reviewer 1 and we have indicated some of the changes above.

Main points:

1/ Outside of MST2 regulations, the authors highlight a couple of molecular regulations fostering ERK reactivation in the resistant cell lines. As indicated by the authors, it is well known that secondary mutation in other oncogenes such as NRAS can reactivate the MAPK pathway. Authors should analyze these mutations in their resistant cell lines, at least in the three RAS isoforms to really confirm that ERK reactivation is, in their experimental conditions, rather due to RTK reactivation. In line with these comments.

We now have data for the mutational status of A375 resistant cells and indeed we could see that in these cells there is a secondary heterozygous mutation NRAS-Q61R (summary shown below and data available upon request). Similar analysis for the other cells lines did not conclusively identify any secondary mutations in these genes and therefore the most likely explanation for the reactivation of RAS is still RTK signalling. The text now reads “Indeed, this might be the case for the A375-R cell lines where we identified a mutation in NRAS, but we could not identify mutations in RAS genes in the other resistant cell lines. A more likely explanation is that RAS activation is caused by the activation of upstream regulators, which causes hyperactivation of several RAS isoforms.”

Cell line	GENE	TRANSCRIT	EXON	c.	p.	Status	Depth	Frequency	Quality parameter.	COSMIC
A375R	NRAS	NM_002524	exon3	c.182A>G	p.Q61R	het	61200	0.5	100	Thyroid_cancer_follicular Epidermal_nevus Non-small_cell_lung_cancer Giant_pigmented_hairy_nevus Neurocutaneous_melanosis Epidermal_nevus_syndrome
A375R	TP53	NM_001126111	exon3	c.98C>G	p.P33R	het	20534	0.55	100	ID=COSM250061;OCCURENCE=1(central_nervous_system),2(upper_aerodigestive_tract),1(urinary_tract),1(liver)
A375	TP53	NM_001126111	exon3	c.98C>G	p.P33R	het	23666	0.48	100	ID=COSM250061;OCCURENCE=1(central_nervous_system),2(upper_aerodigestive_tract),1(urinary_tract),1(liver)
A375	BRAF	NM_004333	exon15	c.1799T>A	p.V600E	hom	87194	1	100	ID=COSM476;OCCURENCE=3(adrenal_gland),3(urinary_tract),4(genital_tract),7(pancreas),16(liver),68(eye),2(prostate),4161(large_intestine),21(biliary_tract),11(breast),4377(skin),9(small_intestine),1(autonomic_ganglia),20(pituitary),242(ovary),32(soft_tissue),7(testis),459(haematopoietic_and_lymphoid_tissue),9534(thyroid),78(lung),1(meninges),12(upper_aerodigestive_tract),717(NS),15(bone),2(oesophagus),4(stomach),228(central_nervous_system)

, authors have only highlighted the stronger activation of RTK signaling but have not commented on the presence of downregulations (some are obvious on the dot blot Fig. 2F)

As also requested by reviewer 1 and 3, we now mention the proteins that show lower phosphorylation in the array.

Finally, it would be interesting to discuss how could be related the mechanisms leading to ERK reactivation and to MST2 downregulation in resistant cells.

In reply to reviewer 1 comments (see above), we have now tested the effect of the inhibition of BRAF and MEK in the resistant cells, and the data show that ERK activation and the protein expression levels of MST2 pathway proteins are not affected. Thus, the reactivation of the ERK pathway in resistant cells does not seem to directly affect the stability of MST2 pathway proteins. This data is now included in the figures and clarified in the text and discussion.

2/ Another conclusion requiring revisions concern the effects of proteasome inhibitors on MST2 stability and their ability to induce cell death. The authors do not present data about the inhibition of the proteasome in parental cell lines. Would MST2 stability also be regulated in a same manner? Authors show that proteasome inhibitors do not lead to apoptosis induction in parental cells. It would be therefore interesting to know if this data is linked to an absence of MST2 stabilization in this context. Otherwise, it would mean that an MST2 target is modified in R cells and would lay upstream of the apoptotic mechanisms. The authors should perform these experiments and comment these results.

We thank the reviewer for this suggestion. As already explained in response to reviewer 1, we do not see significant regulation of MST2 and LATS1 levels in parental cells (figure S3). Therefore, it seems that the degradation of MST2 proteins by the proteasome is part of the acquisition of resistance to BRAFi. We now include this in the text and clarify our interpretation “While in parental cells these inhibitors had little effect in the level of expression of MST2 pathway proteins, both inhibitors consistently and rapidly increased the expression of MST2, LATS1 and YAP1 in all three resistant cell lines (Fig. 3B, Fig. S3).”

3/ IHC data are not all convincing regarding the analyzed sections. The difference of expression of MST2 for the patient 9, 20, 34, 42 are therefore not obvious (supplementary figure 4 and 5). The authors should try to improve this. Also, to be more convincing, they may try to analyze other components of the MST2 pathway by analogy to their analyses performed in vitro (LATS1 and YAP for example) – discuss away as being too much work and limited material.

We respectfully disagree that the sections are not all convincing and as explained in the manuscript two independent histopathologists scored the samples. Below, we include the blind scoring from the second pathologist that is not included in the manuscript for your review. Of note we have reordered the order of the supplementary figures as suggested by reviewer 1.

With respect to the suggestion of extending the number of proteins tested, we fully agree that this is very interesting, but we are very limited by the amount of available material for IHC and consider that is beyond the scope of the current manuscript. We stress in the text that the cohort shown in the manuscript is small and is only indicative, but seems to confirm the relevance of our findings. Of note, we are in the process of validating the antibodies for LATS1/2 and YAP for melanoma sections in our set up and looking for available material to extend our research in a follow up project.

Minor revisions :

1/ "These data indicated that the reduction in MST2, LATS1, and YAP1 proteins expression in PLX4032 resistant cells is due to the acquisition of resistance to the BRAF inhibitors the core proteins of the Hippo pathway are targeted for proteasomal degradation"
This sentence should be reviewed for clarity.

The text has been corrected to "these data indicated that the reduction in MST2, LATS1, and YAP1 proteins expression in PLX4032 resistant cells is due to proteasomal degradation"

2/ Fig4A : - and + are not correctly labelled

This is now corrected.

Reviewer #3 (Comments to the Authors (Required)):

David Romano et al. studied the role of pro-apoptotic MST2 pathway in the acquisition of resistance to BRAFi in melanoma cells. They elegantly show that BRAFi leads to MST2 activation with concomitant increase in cell death. Importantly, this does not occur in in-house resistant cells as these proteins show lower levels of MST2, LATS1 and YAP. This is a consequence of increased ubiquitination, mediated at least in part by MYCBP2 and ITCH, that leads to increased proteasome degradation. Importantly, proteasome inhibition led to increased MST2-dependent cell death in resistant cells. Finally, the authors found MST2 downregulation in biopsies from patient that had progressed to BAFi Vemurafenib, alone or in combination with MEKi. Overall, the paper is very easy

to read, very well written and structured, the experiments performed are adequate and the results, convincing. I've noted some minor text changes to be done and a few issues that I believe can be solved quite easily.

We thank the reviewer for the constructive corrections, and we have now addressed these points in the revised manuscript.

The MST2 pathway is inhibited by mutant BRAF in melanoma cells and the effect is rescued by BRAF inhibitors.

The data presented in figure 1 is well structured and convincing. When citing Fig. 1B, it says "Fig. 1b" Corrected.

Acquisition of resistance to BRAF inhibitors causes the downregulation of MST2 pathway proteins. The data presented in figure 1 is well structured and convincing. However, the authors claim that there's synergy between BRAFi and MEKi in the resistant cells in "Fig. S2C" when it should be "Fig. 2D". Corrected

Looking at the WB, Trametinib alone prevents ERK phosphorylation (except in WM-793 cells, where there's a faint pERK band under Trametinib treatment) so it's impossible to assess whether there's synergy with PLX4032. Please, correct this and/or rephrase it.

The reviewer is correct. We agree that these data do not allow to assess synergy between PLX4032 and trametinib as trametinib already shows full inhibition of ERK activation. We have now corrected the text to better reflect what the figure shows "In parental cells both PLX4032 and trametinib prevent ERK activation, whereas only trametinib inhibited ERK activation in PLX4032 resistant cells (Fig. 2D) "

Also, in Fig. 2F and S2D they show the results from PathScan RTK signaling antibody array. Although the results are clear, the authors don't comment on other possible RTK signalling components that seem to be modulated in resistant cells. For instance (according to the PathScan datasheet): there's increased pSTAT1 in A375R whereas in this cell line IRS1 and Src (both activated in SkMel28R and Src being activated in WM-793) seem to be decreased. Also, S6 Ribosomal Protein shows modulation in the three cell lines though differently: decreased in A375R and SkMel28R but activated in WM-793. Among others, A375 has mutations in CDKN2A and VAV1, SkMel28 in Cdk4, PTEN, EGFR and p53 while WM973 in Cdk4 and PTEN – discuss as also requested by reviewer 2. Maybe their mutational profiles could impact on the signalling rewiring in resistant cells. Please expand on this.

As indicated above in the reply to the other reviewers we have now extended the explanation of the results shown in the array and tested the mutational status which is discussed in the text as indicated in the answers to reviewers 1 and 2.

Also, SkMel28 is incorrectly referred as SkMel20 the first time it appears. Please correct the following: "...Therefore, we assessed the protein expression of LATS1, RASSF1A", it's LATS1 and "...Finally, WM793R had an increase of IGF-1R and the SRC effector SRC...". SRC effector SRC.

Corrected

MST2 pathway proteins are downregulated by proteasome degradation in BRAFi resistant cells. The data presented in figure 3 is well structured and convincing. There is one sentence, though, that it's not clear, at least to me: "These data indicated that the reduction in MST2, LATS1, and YAP1 proteins expression in PLX4032 resistant cells is due to the acquisition of resistance to the BRAF inhibitors the core proteins of the Hippo pathway are targeted for proteasomal degradation." Maybe "after acquisition of resistance to BRAFi, the core proteins of the Hippo pathway are targeted for proteasomal degradation, leading to lower protein levels"

We have corrected this as also mentioned by reviewer 2

Increased ubiquitination promotes MST2 and LATS1 proteasomal degradation in BRAFi resistant cell lines.

The data presented in figure 4 well structured and convincing. In Fig. 4A, the indications for HA and FLAG are missing. Corrected. The author state: "...these E3-ligases might be responsible for the decrease of MST2 and LATS1 ubiquitination in resistant cells". I believe that siRNA-mediated knock-down of MYCBP2 and/or ITCH could confirm this and add value to this finding. I would suggest the authors, if the Editor seems appropriate, to perform these experiments.

We agree that this experiment would be very informative. We tried several times to perform this as part of our original experiments but we could not identify siRNAs that produced a significant downregulation of either MYCBP2 or ITCH. Frustratingly, we have tried again for this review with no success. Therefore, unfortunately at this moment we cannot provide this result. Nevertheless, we still consider that the data shown here, together with the proteomics screening, supports our conclusions.

Proteasome inhibition promotes cell death in MM resistant cell lines though and MST-dependent mechanism.

The data presented in figure 5 well structured and convincing. In figure legend: "Parental and resistant A375, SK-MEL28 and WM-793 were treater with". Please, correct "treater" into "treated". Corrected

Loss of expression of MST2 may correlate with development of resistance to BRAFi in patients
The data presented in figure 6 well structured and convincing. "...we used a small cohort of nine human metastatic melanoma patients who were resistant to vemurafenib treatment alone or in combination with MEK inhibitors, and where both pre- treatment...". Please, correct this "and where both". Maybe "from whom" Both corrected

Discussion

In the third paragraph, the authors state: "the phosphorylation of the major MYCBP3 phosphorylation site, serine 3503, is downregulated by both PLX4032 as well as the MEK inhibitor AZD6244 (selumetinib) (Stuart et al., 2015)". I believe there's a mistake there since MYCBP3 doesn't exist and I couldn't find it in Stuart et al. paper. MYCBP2 phosphorylation sites don't include Ser 3503 but 3505. Please, correct and/or clarify this.

Apologies for the typos, the reviewer is correct and we have now included the right information in the text.

August 1, 2022

RE: Life Science Alliance Manuscript #LSA-2022-01445-TR

Dr. David Matallanas
University College Dublin
Systems Biology Ireland
Belfield
Dublin
Ireland

Dear Dr. Matallanas,

Thank you for submitting your revised manuscript entitled "Proteasomal downregulation of the MST2 pathway contributes to BRAF inhibitor resistance in melanoma". We would be happy to publish your paper in Life Science Alliance pending final revisions necessary to meet our formatting guidelines.

- please address Reviewer 1 and 2's remaining comments
- please upload your supplementary figures as single files
- please add ORCID ID for secondary corresponding author-they should have received instructions on how to do so
- please use the [10 author names, et al.] format in your references (i.e. limit the author names to the first 10)
- please double-check your label for Figure S5 in the figure file (it is currently labeled as Figure S4)

Figure Check:

- scale bars are needed for Figure S5, Figure S6
- please include sizes next to all blots

A. FINAL FILES:

B. MANUSCRIPT ORGANIZATION AND FORMATTING:

Sincerely,

Reviewer #1 (Comments to the Authors (Required)):

The manuscript by Romano et al. convincingly demonstrates a new resistance mechanism to Vemurafenib involving the downregulation of the pro-apoptotic MST2 pathway. The authors have performed many new experiments during the revision that confirm their findings and improve the manuscript. All of my suggestions or concerns were addressed with additional experiments. I will go through all of my five comments in the following. In my opinion, my second point concerning the use of different BRAF inhibitors needs further clarification. All the other questions or comments have been answered or corrected perfectly.

1) The authors have tried to analyze if the inhibitory interaction between BRAF and MST2 persists in the resistant cells. I understand the technical difficulties of these endogenous IPs, where only so little of MST2 is expressed. I wonder if performing the reverse IP, precipitating MST2, and looking for BRAF interaction would have delivered a clearer answer (as you would have been able to directly compare MST2 levels on one SDS Gel). But probably the authors have already tried this. Nevertheless, the BRAF IP shown in the new Fig. S2E indicates - as the authors state correctly - that no interaction is detected, which might be due to the low levels of MST2.

I appreciate that the authors analyzed the MST2 signaling status in the BRAFi and MEKi inhibitor treated cells to see whether the reduction of ERK pathway activity can reverse the downregulation of the MST2-LATS2 pathway components (Fig. S2D). It has been clearly shown now that the downregulation is ERK independent.

2) The authors have now included a new Supplementary Fig. (S1) analyzing the effects of different BRAF inhibitors. The authors state that "All of them decreased MST2-BRAF interaction" and point out in the discussion that MST2 activation might be a common mechanism of action of these inhibitors. However, the reduction that is very pronounced with Vemurafenib (about 90-100% reduction according to Fig. 1C) is only marginal with the other inhibitors (between 30% and 10 %, new Fig. S1). I would at least suggest, that the authors rephrase the sentence to "All of them decreased MST2-BRAF interaction, but to a lesser extent." Also the Figure legend has to be corrected. Is it 1 hour or 24 hours of treatment? Did you mean 2 μ M Encorafenib instead of 2 mM? As pERK levels are only mildly affected by the drugs it could be that the inhibition was not efficient enough to see an effect on the interaction. I suggest using at least 10 nM instead of 5 nM Dabrafenib (10.3892/or.2017.5963), 5-10 μ M Sorafenib, 0.5 -1 μ M Encorafenib and at least 80 nM TAK-632 (10.1038/s41467-020-18123-2). If the experiment will not be repeated, I would like to ask for a more careful description of the data.

3) Thank you for including the "control IP" in the Supplement (new Fig. S4B)

4) Importantly, the authors can now show that proteasome inhibition specifically upregulates LATS1 and MST2 in resistant cells and not in parental cells (new Fig. S3), indicating that the acquired changes in proteasomal degradation are specific to the resistant cancer cells. Similarly, the authors now show that Bortezomib treatment specifically induces PARP cleavage in resistant cells at various concentrations while parental cells are not affected (new Fig. S3).

5) Thank you for reorganizing Fig. S5 (mistakenly named S4) and S6 and for improving text clarity. I think there still is a mistake in Figure legend 6. Patient 6 and 13 were (according to Table 1) treated with the combination. It should read: "The figure shows expression of MST2 in patients 6 and 13 before treatment (pre-treatment) and after treatment with BRAF (dabrafenib) and MEK (inhibitors (BRAFi))."

Reviewer #2 (Comments to the Authors (Required)):

David Romano et al. studied the role of pro-apoptotic MST2 pathway in the acquisition of resistance to BRAFi in melanoma cells. They first demonstrate in melanoma cell lines that MST2 activity is buffered through BRAFv600 interaction, leading to MST2 reactivation in presence of inhibitors of BRAFv600. Then, by comparing parental cell lines and their resistant counterparts engineered in vitro, they highlight that acquired resistance to BRAFv600 inhibitors is correlated to the destabilization of MST2 through an ubiquitin-proteasome dependent mechanism. The main conclusions are convincingly supported by the data and this study open a new way to treat patients with melanoma resistant to BRAFi.

Authors reply to my comments and the paper is now acceptable for publication. However, I am still not fully convinced by the interpretation of IHC data. The data are clear for a couple of patients but for others, the difference might just be a trend, especially with a measurement not quantitative such as IHC. I would thus modify this sentence "lost or reduced in 8 out the 9 patients" to be less affirmative. Also, it seems that microscope magnification 40X for the patient 35 pre-treatment is incorrectly labelled since the size of the cells on this slide is strongly different from the others.

Reviewer #3 (Comments to the Authors (Required)):

I thank the authors for addressing all the points I had noted. I believe the paper requires no further changes now.

Editor's requests:

- please address Reviewer 1 and 2's remaining comments. Done
- please upload your supplementary figures as single files. Done
- please add ORCID ID for secondary corresponding author-they should have received instructions on how to do so. Done if it is not showing in the system it is 0000000157775016
- please use the [10 author names, et al.] format in your references (i.e. limit the author names to the first 10). Done
- please double-check your label for Figure S5 in the figure file (it is currently labeled as Figure S4). Done

Figure Check:

- scale bars are needed for Figure S5, Figure S6. Done
- please include sizes next to all blots. Done, please note that for aesthetic reasons in figures 1C and 4A the MW is shown beside the blots on the left but are the same for the panel.

Reviewer #1 (Comments to the Authors (Required)):

The manuscript by Romano et al. convincingly demonstrates a new resistance mechanism to Vemurafenib involving the downregulation of the pro-apoptotic MST2 pathway. The authors have performed many new experiments during the revision that confirm their findings and improve the manuscript. All of my suggestions or concerns were addressed with additional experiments. I will go through all of my five comments in the following. In my opinion, my second point concerning the use of different BRAF inhibitors needs further clarification. Please see below to see the clarifications done. All the other questions or comments have been answered or corrected perfectly.

1) The authors have tried to analyze if the inhibitory interaction between BRAF and MST2 persists in the resistant cells. I understand the technical difficulties of these endogenous IPs, where only so little of MST2 is expressed. I wonder if performing the reverse IP, precipitating MST2, and looking for BRAF interaction would have delivered a clearer answer (as you would have been able to directly compare MST2 levels on one SDS Gel). But probably the authors have already tried this. Nevertheless, the BRAF IP shown in the new Fig. S2E indicates - as the authors state correctly - that no interaction is detected, which might be due to the low levels of MST2. Indeed, we have done the IP both ways as mentioned by the reviewer with the same lack of clarity.

I appreciate that the authors analyzed the MST2 signaling status in the BRAFi and MEKi inhibitor treated cells to see whether the reduction of ERK pathway activity can reverse the downregulation of the MST2-LATS2 pathway components (Fig. S2D). It has been clearly shown now that the downregulation is ERK independent.

2) The authors have now included a new Supplementary Fig. (S1) analyzing the effects of different BRAF inhibitors. The authors state that "All of them decreased MST2-BRAF interaction" and point out in the discussion that MST2 activation might be a common mechanism of action of these inhibitors. However, the reduction that is very pronounced with Vemurafenib (about 90-100% reduction according to Fig. 1C) is only marginal with the other inhibitors (between 30% and 10 %, new Fig. S1).

I would at least suggest, that the authors rephrase the sentence to "All of them decreased

MST2-BRAF interaction, but to a lesser extent." We now have changed the text as suggested. Also the Figure legend has to be corrected. Is it 1 hour or 24 hours of treatment? It is 24h as indicated in supp figure 1A. This is in line with in figure 1A/C where we used 1h and 24h. For simplicity in, as there were several inhibitors, in the experiment shown in supp Figure 1A we decided to perform 24h treatment alone. Figure S1B is 1 and 4 hours as indicated, since we have seen that the interaction between LATS1 and MST2 upon proapoptotic stimuli increase at shorter time points (Matallanas et al 2007 and Romano et al 2013) Did you mean 2 μ M Encorafenib instead of 2 mM? Apologies for the typo this is corrected now. As pERK levels are only mildly affected by the drugs it could be that the inhibition was not efficient enough to see an effect on the interaction. I suggest using at least 10 nM instead of 5 nM Dabrafenib (10.3892/or.2017.5963), 5-10 μ M Sorafenib, 0.5 -1 μ M Encorafenib and at least 80 nM TAK-632 (10.1038/s41467-020-18123-2). If the experiment will not be repeated, I would like to ask for a more careful description of the data. We do not consider that repeating this experiment is necessary to support our conclusion and we have done the changes suggested by the reviewer and corrected the typos. The text now read "All of them decreased the MST2-BRAF interaction 24 hour post-treatment, but to a lesser extend"

3) Thank you for including the "control IP" in the Supplement (new Fig. S4B)

4) Importantly, the authors can now show that proteasome inhibition specifically upregulates LATS1 and MST2 in resistant cells and not in parental cells (new Fig. S3), indicating that the acquired changes in proteasomal degradation are specific to the resistant cancer cells. Similarly, the authors now show that Bortezomib treatment specifically induces PARP cleavage in resistant cells at various concentrations while parental cells are not affected (new Fig. S3).

5) Thank you for reorganizing Fig. S5 (mistakenly named S4) and S6 and for improving text clarity. I think there still is a mistake in Figure legend 6. Patient 6 and 13 were (according to Table 1) treated with the combination. It should read: "The figure shows expression of MST2 in patients 6 and 13 before treatment (pre-treatment) and after treatment with BRAF (dabrafenib) and MEK (inhibitors (BRAFi))." We now made the change to the text suggested by the reviewer.

Reviewer #2 (Comments to the Authors (Required)):

David Romano et al. studied the role of pro-apoptotic MST2 pathway in the acquisition of resistance to BRAFi in melanoma cells. They first demonstrate in melanoma cell lines that MST2 activity is buffered through BRAFv600 interaction, leading to MST2 reactivation in presence of inhibitors of BRAFv600. Then, by comparing parental cell lines and their resistant counterparts engineered in vitro, they highlight that acquired resistance to BRAFv600 inhibitors is correlated to the destabilization of MST2 through an ubiquitin-proteasome dependent mechanism. The main conclusions are convincingly supported by the data and this study open a new way to treat patients with melanoma resistant to BRAFi. Authors reply to my comments and the paper is now acceptable for publication. However, I am still not fully convinced by the interpretation of IHC data. The data are clear for a couple of patients but for others, the difference might just be a trend, especially with a measurement not quantitative such as IHC. I would thus modify this sentence "lost or reduced in 8 out the 9 patients" to be less affirmative. We do disagree with the reviewer that there is a need to change the text. As we have already pointed out in our first rebuttal, two pathologists

consider that the data is of necessary quality for scoring the changes in a similar way as they do for many other histology biomarkers used for clinical diagnosis. We actually clearly indicate that we are cautious in the interpretation of our results in the next sentence "Although the sample size is small, these data **suggest** that a reduction in MST2 expression is part of the mechanism how melanomas develop resistance to BRAFi in patients". Nevertheless, we have changed to "seems to be lost or reduce" which we think is even less assertive. Also, it seems that microscope magnification 40X for the patient 35 pre-treatment is incorrectly labelled since the size of the cells on this slide is strongly different from the others. Thanks for detecting this oversight. This is correct and We have now changed the figures to show the correct magnification.

Reviewer #3 (Comments to the Authors (Required)):

I thank the authors for addressing all the points I had noted. I believe the paper requires no further changes now.

August 19, 2022

RE: Life Science Alliance Manuscript #LSA-2022-01445-TRR

Dr. David Matallanas
University College Dublin
Systems Biology Ireland
Belfield
Dublin, Ireland

Dear Dr. Matallanas,

Thank you for submitting your Research Article entitled "Proteasomal downregulation of the MST2 pathway contributes to BRAF inhibitor resistance in melanoma". It is a pleasure to let you know that your manuscript is now accepted for publication in Life Science Alliance. Congratulations on this interesting work.

DISTRIBUTION OF MATERIALS:

Again, congratulations on a very nice paper. I hope you found the review process to be constructive and are pleased with how the manuscript was handled editorially. We look forward to future exciting submissions from your lab.

Sincerely,
